**The Aare main overdeepening on the northern margin of the European Alps: Basins, riegels,**
**and slot canyons**
Fritz Schlunegger[1], Edi Kissling[2], Dimitri Bandou[1,3], Guilhem Douillet[1], David Mair[1], Urs Marti[4],
Regina Reber[1], Patrick Schläfli[1,5], and Michael Schwenk[1,6]
[1]Institute of Geological Sciences, University of Bern, Baltzerstrasse 1+3, 3012 Bern, Switzerland
[2]Department of Earth Sciences, ETH Zürich, Sonneggstrasse 5, 8092 Zürich, Switzerland
[3]Department of Environmental Sciences, University of Virginia, 291 McCormick Rd., Charlottesville,
VA 22904-4123, USA
[4]Landesgeologie Swisstopo, Seftigenstrasse 264, Postfach, 3084 Wabern, Switzerland
[5]Institute of Plant Sciences and Oeschger Centre for Climate Change Research, Altenbergrain 21,
3013 Bern, Switzerland
[6]Bayerisches Landesamt für Umwelt, Umweltdienstleistungen, Hof, 95030 Hof Saale, Germany
fritz.schlunegger@unibe.ch
**Abstract**
This work summarizes the results of an interdisciplinary project where we aimed to explore the origin
of overdeepenings through a combination of a gravimetry survey, drillings and dating. To this end, we
focused on the Bern area, Switzerland, situated on the northern margin of the European Alps. This area
experienced multiple advances of piedmont glaciers during the Quaternary glaciations, resulting in the
carving of the main overdeepening of the Aare River valley (referred to as the Aare main
overdeepening). This bedrock depression is tens of km long and up to several hundreds of m to a few
km wide. We found that in the Bern area, the Aare main overdeepening is made up of two >200 m-deep
troughs that are separated by a c. 5 km-long and up to 150 m-high transverse rocky ridge, interpreted
as a riegel. The basins and the riegel are overlain by a >200 m- and a c. 100 m-thick succession of
Quaternary sediments, respectively. The bedrock itself is made up of a Late Oligocene to Early Miocene
suite of consolidated clastic deposits, which are part of the Molasse foreland basin. In contrast, the
Quaternary suite comprises a middle Pleistocene to Holocene succession of unconsolidated glacio-
lacustrine gravel, sand and mud. A synthesis of published gravimetry data revealed that the upstream
stoss side of the bedrock riegel is c. 50% flatter than the downstream lee side. In addition, information
from >100 deep drillings reaching depths >50 m suggests that the bedrock riegel is dissected by an
anastomosing network of slot canyons. Apparently, the slot canyons established the hydrological
connection between the upstream and downstream basins during their formation. Based on published
modelling results, we interpret that the riegels and canyons were formed through incision of subglacial

meltwater during a glacier's decay state, when large volumes of meltwater were released. It appears that such a situation has repeatedly occurred since the Middle Pleistocene Transition approximately 800 ka ago, when large, several hundreds of m-thick and erosive piedmont glaciers began to advance far into the foreland. This resulted in the deep carving of the inner-Alpine valleys and additionally in the formation of overdeepenings, riegels and slot canyons on the plateau situated on the northern margin of the Alps.

## 1      Introduction

Overdeepenings are bedrock depressions below the current fluvial base-level (e.g., Jørgensen and Sandersen, 2006; Dürst Stucki et al., 2010; 2013; Fischer and Häberli, 2012). The downstream closures of these basins have adverse slopes that generally dip in the upstream direction (Häberli et al., 2016). Because bedrock depressions with such characteristics (Figure 1) are commonly found in previously glaciated areas, their formation has been interpreted as resulting from the erosional work of glaciers and/or subglacial meltwater (Wrigth, 1973; Herman and Braun, 2008; Egholm et al., 2009; Kehew et al., 2012; Patton et al., 2016; Liebl et al., 2023; and many others). Overdeepenings have been reported for the Quaternary from beneath the Greenland and Antarctic glaciers (Ross et al., 2011; Patton et al., 2016), but also in the North Sea (Moreau et al., 2012, Lohrberg et al., 2022), North America (Wright, 1973; Lloyd et al., 2023) and northern Europe including Scandinavia (Clark and Walder, 1994; Piotrowski, 1997; Krohn et al., 2009). Glaciogenic paleovalleys are not only limited to the Quaternary but were also described for Paleozoic glaciations (e.g. Douillet et al., 2012; Dietrich et al., 2021). In the European Alps, such erosional troughs occur in Alpine valleys as well as on foreland plateaus on either side of this mountain belt (Preusser et al., 2010; Dürst Stucki and Schlunegger, 2013; Magrani et al., 2020). Pollen analysis (Welten, 1982; 1988; Schlüchter, 1989; Schläfli et al., 2021), dating using optically stimulating luminescence methods (Preusser et al., 2005; Dehnert et al., 2012; Büchi et al., 2018; Schwenk et al., 2022a) and $^{14}$C ages established on organic matter encountered in the overdeepening fill (Kellerhals and Häfeli, 1984) showed that these troughs were formed after the Middle Pleistocene Transition, which occurred c. 800 ka ago (Schlüchter, 2004). Geophysical surveys (e.g., Rosselli and Raymond, 2003; Reitner et al., 2010; Stewart and Lonergan, 2011; Stewart et al., 2013; Perrouty et al., 2015; Burschil et al., 2018; 2019; Ottesen et al., 2020) in combination with drillings (Jordan, 2010; Dürst Stucki et al., 2010; Büchi et al., 2017; 2018; Gegg et al., 2021; Bandou et al., 2022; 2023; Anselmetti et al., 2022; Schwenk et al., 2022a, b; Gegg and Preusser, 2023; Schaller et al., 2023; Schuster et al., 2024) disclosed that such overdeepenings can be several km wide, tens of km long and >200 m deep. The surveys also showed that overdeepenings are typically composed of individual sub-basins, separated by bedrock swells or bumps oriented transverse to the flow of a former glacier, hereafter called riegels (Cook and Swift, 2012), yet the specific details of such a geometry have not yet been elaborated.

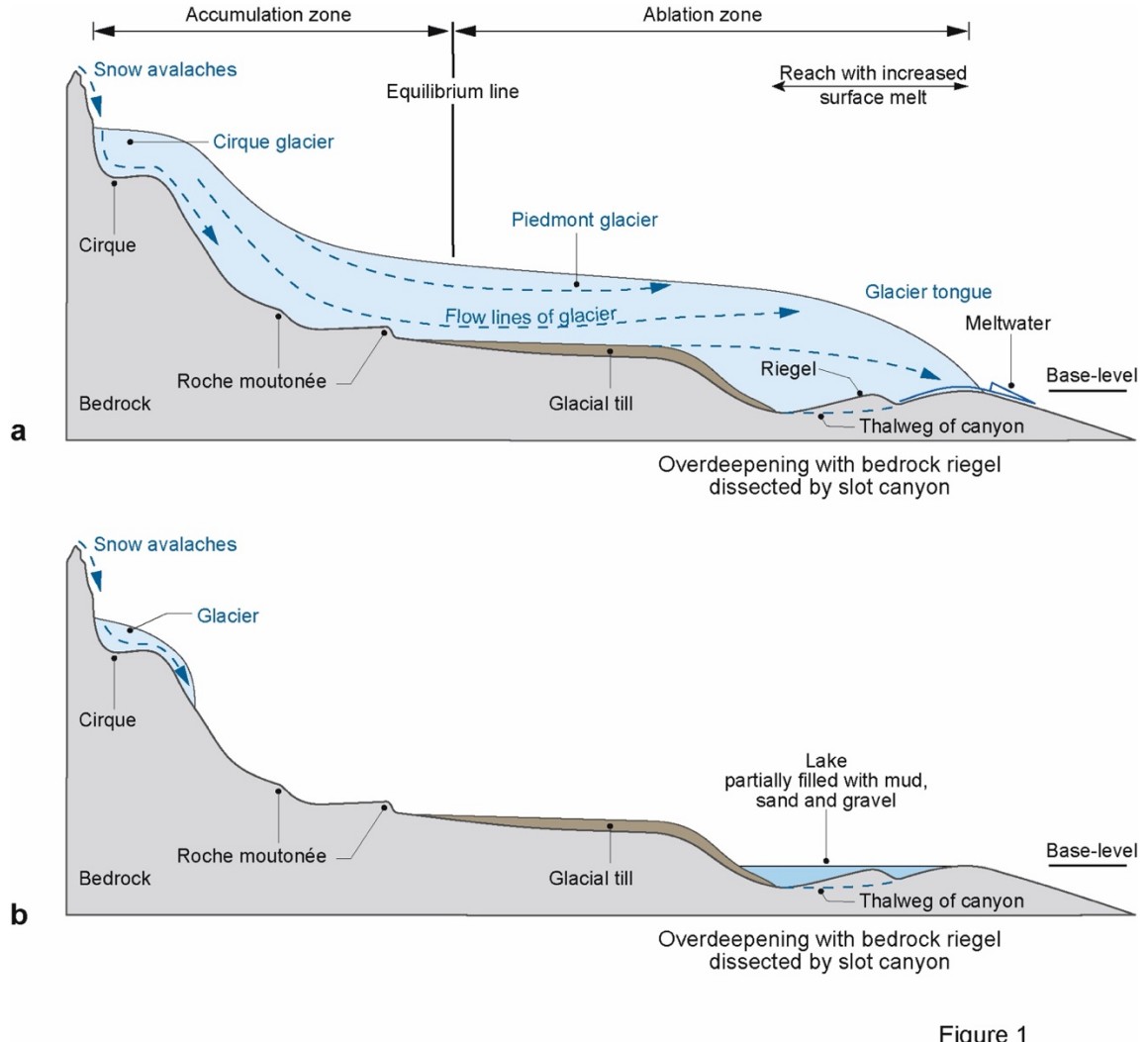


Figure 1

Figure 1:    Architecture of a landscape sculpted by piedmont glaciers during glaciations. a) Situation immediately following a full glacial period during which a piedmont glacier, which extended far into the foreland, started to melt. As a result, large volumes of meltwater are produced in the ablation zone close to the glacier's tongue. This meltwater has the potential to contribute to the erosional downwearing of the bedrock, and it can cause the incision of canyons into bedrock riegels, which separate two overdeepened basins. b) During interglacial time periods, the piedmont glaciers disappear, and small ice caps may be preserved in the higher parts of a mountain belt. During this time, the overdeepened basin will be filled by lacustrine sediments and/or will eventually host a lake. Modified after Schlunegger and Garefalakis (2023).


Here, we summarize the results of an interdisciplinary project where we aim to explore the origin of
overdeepenings using a combination of data collected through a gravimetry survey (Bandou, 2023a;
Bandou et al., 2022, 2023), drillings (Reber and Schlunegger, 2016; Schwenk et al., 2022a, b) and
dating (Schläfli et al., 2021; Schwenk et al., 2022a). We focus our study on the Bern area situated on
the northern margin of the European Alps (Figure 2). For this region, we draw a map of the bedrock
topography combining the results of a gravimetry survey in the region (Bandou, 2023a; Bandou et al.,
2023) with information obtained through drillings. This map shows that an overdeepened trough or a
tunnel valley system, referred to as the Aare main overdeepening (Schwenk et al., 2022a), is made up
of two basins separated by a bedrock riegel, which itself is cut by one or multiple slot canyons. This
structure has a similar geometry as many examples reported from formerly glaciated landscapes
(Brocklehurst and Whipple, 2002; Brocklehurst et al., 2008) and particularly from Alpine valleys,
which points to similar processes resulting in their formation.

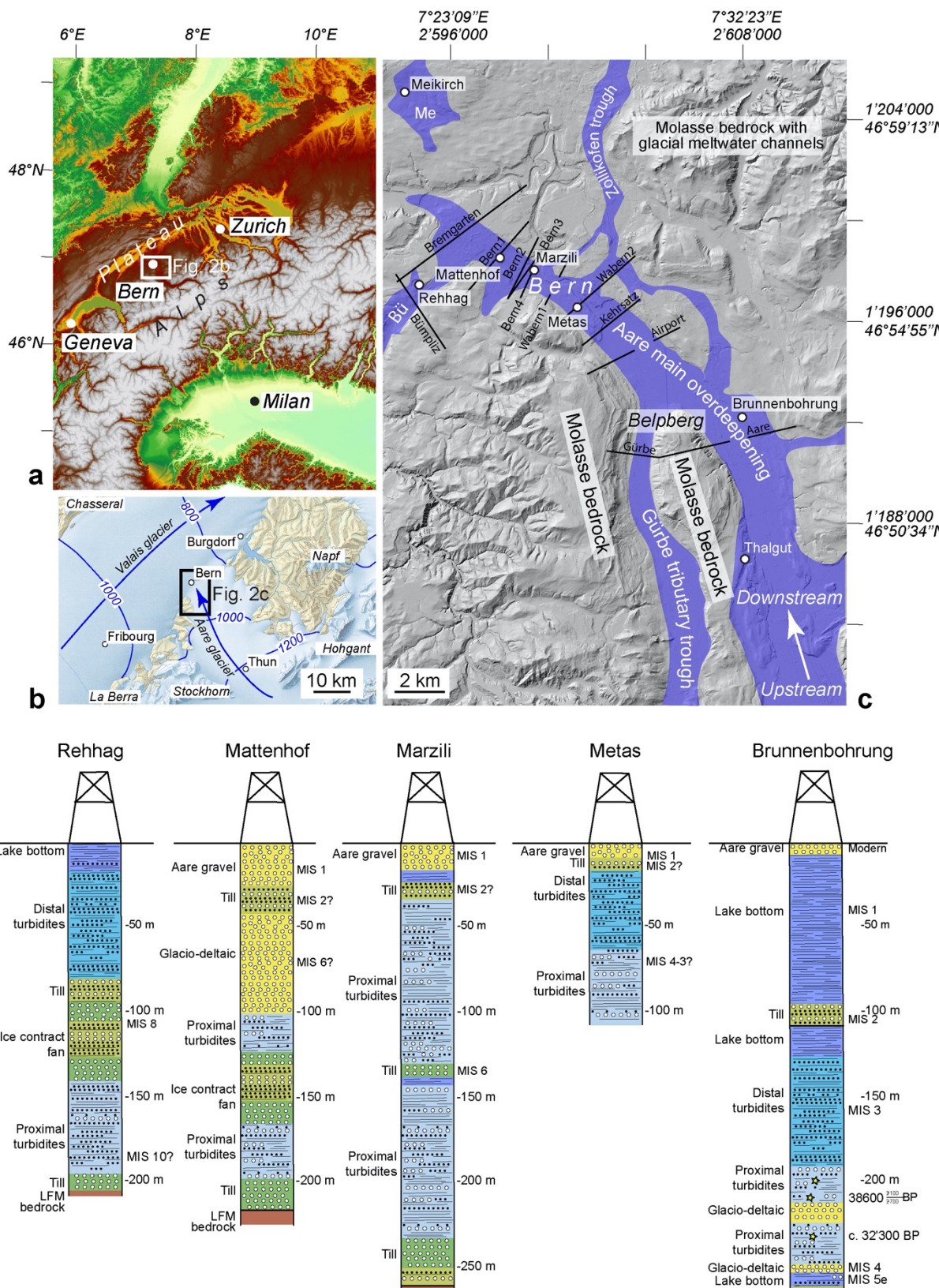

Figure 2


Figure 2: Local setting illustrating the a) Alpine arc (modified from Bandou et al., 2023) with latitudes and longitudes, b) the study area during the Last Glacial Maximum (LGM; map with isohypses of the glacier's surfaces taken from Bini et al., 2009), c) the surface geomorphology (2 m-SwissAlti3D DEM © swisstopo) together with the orientation of the Aare main overdeepening, taken from Reber and Schlunegger (2016), and d) information from drillings. The figure c) shows (i) the sections along which gravity data was collected (black lines; Bandou et al., 2022; 2023), and (ii) the sites (white circles) where sediments in drillings (Rehhag: Schwenk et al., 2022a, b; Meikirch: Welten, 1982; Preusser et al., 2005; Schläfli et al., 2021; Brunnenbohrung: Kellerhals and Häfeli, 1984; Zwahlen et al., 2021) and exposures (Thalgut: Welten, 1982; 1988; Schlüchter, 1989; Preusser and Schlüchter, 2004) were either dated with various techniques, or where existing ages were reconfirmed by a subsequent analysis. Me=Meikirch overdeepening; Bü=Bümpliz trough. The numbers along the figure margin refer to the Swiss coordinate system (CH1903+) and are complemented with information on latitudes and longitudes. Panel d) presents the logs of key drillings. The logs of the Brunnenbohrung (modified after Kellerhals and Häfeli, 1984) and Mattenhof drillings (modified after Geotest, 2013) were reconstructed from cuttings; the material at Metas and Rehhag was cored (Geotest, 1997; Schwenk et al., 2022a), whereas the sedimentary log of the Marzili drilling is based on a combination of cuttings and gamma ray data (Gees, 1974). The age models of the sequences encountered in the Mattenhof, Marzili, Metas and Brunnenbohrung drillings were based on regional correlations with dated horizons (see Bandou et al., 2023, for further information).

## 2 Setting

### 2.1 *Overdeepened troughs in the Bern area*

The target overdeepening near Bern was sculpted by the Aare piedmont glacier with sources in the Central European Alps. From there, the Aare glacier flowed onto the Swiss Plateau (Figure 2a) over a distance of >20 km, and it merged with the Valais glacier north of Bern, at least during the Last Glacial Maximum (LGM) c. 20 ka ago (Figure 2b). Upstream of the city area of Bern, two bedrock depressions, referred to as the Gürbe tributary trough and the Aare main overdeepening (Figure 2c), form prominent basins. They are between c. 150 (Gürbe trough; Geotest, 1995) and >250 m deep (Aare main trough, Kellerhals and Häfeli, 1984), and several km wide (Bandou et al., 2022). Downstream of the city of Bern, the Aare main overdeepening splits into several distributary branches. Among these, the Bümpliz trough ('Bü' in Figure 2c) is the most prominent one with a depth >200 m (Schwenk et al., 2022a, b). The other depressions such as the Zollikofen trough are shallower and reach a depth of <150 m (Reber and Schlunegger, 2016). The study region also hosts the Meikirch overdeepening (labelled as 'Me' on Figure 3c), a nearly 200 m-deep trough (Dürst Stucki et al., 2010; Dürst Stucki and Schlunegger, 2003), which appears to be isolated from the rest of the overdeepening system (Reber and Schlunegger, 2016). Although the area between the northern termination of the Aare main overdeepening and the Meikirch trough is made up of exposed bedrock (Gerber, 1927), the possibility of a connection between both depressions via a narrow canyon, while quite unlikely according to Reber and Schlunegger (2016), cannot be completely ruled out. The Aare main overdeepening itself is the most prominent trough in the city area of Bern and has a maximum depth of nearly 250 m (Reber and Schlunegger, 2016).

 *2.2    Chronologic framework of overdeepening fill*

The Quaternary fill of the Aare main overdeepening has been placed into the chronological framework
of glacial advances onto the Swiss plateau during the past glaciations by previous authors. South of
Bern, the Thalgut section (Figure 2c) disclosed the occurrence of pollen fragments embedded in a
lacustrine sequence at the base and near the top of the section (Schlüchter, 1989). The pollen assemblage
at the base was assigned to the Holsteinian interglacial period (Welten, 1982; 1988; Schlüchter, 1989;
Preusser and Schlüchter, 2004), which either corresponds to MIS 9 (Roger et al., 1999) or MIS 11 (see
discussion in Preusser et al., 2011; Koutsodendris et al., 2012; and Schwenk et al., 2022a for a
discussion of ages). The lacustrine sediments near the top of the same suite were assigned to MIS 5e
(Welten, 1982; 1988; Schlüchter, 1989). Approximately 6 km farther downstream of the Thalgut
section, the Brunnenbohrung drilling (Figure 2d) penetrated nearly the entire sedimentary sequence of
the Aare main overdeepening. Based on lithostratigraphic constraints and $^{14}$C ages established on
organic fragments, Kellerhals and Häfeli (1984) and subsequently Zwahlen et al. (2021) assigned an
age postdating MIS 6 to the entire succession. Farther north of Bern, Schwenk et al. (2022a) used the
results of feldspar luminescence dating to propose that the sedimentary suite penetrated by the Rehhag
drilling has an age of MIS 8 and older (Figure 2d). Finally, the Neubrügg section, which is exposed at
the NE end of the Bremgarten profile (Figure 2a), exposes c. 60-70 m-thick sequence with a till at the
base and the top. The succession also includes sand and gravel deposits with pollen fragments between
the till layers, which may indicate the end of a warm period according to Lüthy et al. (1963). Bandou
et al. (2023) used this information to suggest that the till at the base and the top of the suite could
correspond to the MIS 6 and MIS 2 glaciations, respectively, while the sediments recording a warm
period (or the end of a warm time interval) could have been deposited during MIS 5e. We acknowledge
that all of the aforementioned ages are not precise enough to reconstruct in detail the history of how and
when the overdeepenings were formed, but they are consistent with the view that the deep troughs in
the Bern area were originally formed after the Middle Pleistocene Transition c. 800 ka ago (Schlüchter,
2004) and thus during the same period when the U-shaped Alpine valleys were carved (Häuselmann et
al., 2007; Valla et al., 2011).

*2.3    Lithological architecture of bedrock*
The bedrock in the region comprises an amalgamated suite of Early Miocene Upper Marine Molasse
(UMM) sandstone beds south of Bern. Sedimentological analyses showed that these sediments were
deposited in a shallow marine, mostly coastal environment (Garefalakis and Schlunegger, 2019). In the
region north of Bern, the bedrock is made up of a Late Oligocene to Early Miocene suite of Lower
Freshwater Molasse (LFM) sandstones and mudstones (Isenschmid, 2019). These sediments were
deposited in a fluvial environment comprising channel fills and floodplains made up of sandstones and
mudstones, respectively (Platt and Keller, 1992; Isenschmid, 2019). The bedding of the Molasse
sediments and the contact between the UMM and the LFM gently dips towards the south by c. 10°
(Isenschmid, 2019), with the consequence that south of Bern, the base of the Aare main overdeepening
often consists of LFM deposits, while most of the upper part of the overdeepening is laterally bordered
by bedrock made up of UMM. In addition, it has been postulated that the UMM sediments have a lower
erodibility than the underlying LFM unit, based on the observation that the UMM forms a cap rock in
the region (Isenschmid, 2019). Finally, Isenschmid (2019) documented that the Molasse bedrock
beneath the Bern city area is dissected by left-lateral faults that strike NW-SE, offering zones of
mechanical weaknesses.

*2.4     Lithological architecture of overdeepening fill*
Schwenk et al. (2022a) grouped the Quaternary sediments recovered from the Rehhag drilling into
distinct facies assemblages based on a detailed description of the 210 m-long drill core. The first
assemblage, interpreted as subglacial traction till and encountered at the base of the Rehhag sequence
(Figure 2d), comprises a suite of gravel with angular to rounded clasts that are embedded in a sandy to
silty, strongly compacted and sheared matrix. This element shows strong lithologic similarities to the
second facies assemblage, which consists of an alternation of gravel and sand layers and which was
encountered in the middle of the drill core. This assemblage was interpreted by Schwenk et al. (2022a)
as ice contact fan deposits. Finer-grained facies assemblages consist either of (i) sand layers with mud
and gravel interbeds (in the section between c. 195 and 140 m depth), interpreted as deposits from
proximal turbidity currents, or of (ii) alternating sand and mud layers (in the section between c. 80 and
20 m depth), representing deposits from distal turbidity currents (Figure 2d). The uppermost sequences
made up of mud layers with isolated clasts (drop stones) were interpreted as lake bottom sediments
(Figure 2d). Based on OSL dating, Schwenk et al. (2022a) assigned a MIS 8 or possibly older age to
the sequence at the Rehhag.
At the Mattenhof situated farther to the ENE, the log of the >200 m-thick Quaternary sequence was
reconstructed based on cuttings (Geotest, 2013). The suite starts with a c. 20 m-thick gravel, which is
overlain by a c. 30 m-thick succession of mud with gravel interbeds. Following the scheme of Schwenk
et al. (2022a), we interpret this sequence as a till that is overlain by material supplied by turbidites
(Figure 2d). The following sequence between 166 and 124 m drilling depth comprises gravel beds with
mud and interbedded sand layers. This represents a more proximal facies than the underlying sequence
and could, according to the interpretation scheme of Schwenk et al. (2022a), correspond to ice-contact
fan deposits. The overlying suite, up to a depth of 102 m, is made up of mud with some gravel layers
and isolated clasts. Similarly to the basal unit, this material was most likely supplied by turbidity
currents. The isolated clasts in this suite could represent drop stones. The upper part of the Mattenhof
section consists of gravel deposits up to a depth of 42 m, followed by a silty gravel unit between 42 and
28 m depth, and then another gravel sequence reaching the top of the section. This gravelly suite could

potentially represent a glacio-deltaic system, postdating MIS 6 according to the regional correlation of Bandou et al. (2023).

For the Marzili drilling, information about the stratigraphic architecture of the >250 m-thick Quaternary suite was reconstructed based on cutting and gamma ray data (Gees, 1974). There, the sequence starts with suite made up of gravels and interbedded sand layers, which we interpret as a till or as ice contact fan deposits following the interpretation scheme of Schwenk et al. (2022a). These deposits are overlain by a sequence of mud with interbedded gravel layers, possibly representing an environment where a large portion of the material was supplied by turbidity currents. A 4 m-thick gravel unit was encountered at a drilling depth of c. 130, which could represent a till. The overlying sequence comprises an alternation of gravel and mud (Gees, 1974), possibly representing a suite of sediments supplied by turbidity currents. Towards the top, the Marzili section comprises a 6 m-thick sequence made up of mud, and it ends with a 20 m-thick fluvial gravel. Based on regional correlations, Bandou et al. (2023) tentatively assigned a post-MIS 6 age to the sequence overlying the gravel at the depth of 130 m.

Farther north, the Metas drilling penetrated a 110 m-thick sequence without reaching the bedrock (Geotest, 1997). The drilled core starts with a c. 90 m-thick suite made up of mud and sand layers, which contains isolated clasts. These sediments were most likely deposited in a proglacial lake by turbidity currents. In this context, the isolated clasts could represent drop stones (Schwenk et al., 2022a). This sequence is overlain by a till (MIS 2?) and finally by a c. 15 m-thick proglacial gravel (Geotest, 1997). Finally, south of Bern, the >250 m-thick succession at the Brunnenbohrung site (log based on cuttings) starts with a few m-thick till (possibly MIS 6), yet the drilling did not reach the bedrock (Kellerhals and Häfeli, 1984). The till is overlain by a several m-thick alternation of mud, silt and sand layers (possibly MIS 5e). The latter unit is then followed by a >30 m-thick suite made up of a glacio-deltaic gravel, alternations of gravel, mud and sand, and then again by a 10 m-thick gravel. It continues with a fining-upward sequence deposited by turbidity currents at the bottom of a lake. Measurements of [14]C concentrations in organic matter point to an age of MIS 3 (Kellerhals and Häfeli, 1984). The topmost 100 m-thick suite starts with a till at a depth of c. 100 m (possibly MIS 2), which grades into a fining-up sequence made up of mud and silt deposited at the bottom of a lake. The Brunnenbohrung section ends with a fluvial gravel.

In summary, the Quaternary successions are spatially highly heterogeneous as disclosed by the drillings, but they all record the same depositional setting as the sediments were most likely deposited in a glacio-lacustrine environment (e.g., Schwenk et al., 2022a). Apparently, the material supply was spatially highly heterogeneous (Schwenk et al., 2022b) as evidenced by the varying locations where coarse-grained facies assemblages were encountered in the drillings (Figure 2d).

*2.5 Density of Molasse bedrock and Quaternary sediments*

Data on the bulk density of the Molasse bedrock and the overlying Quaternary sediments is crucial for interpreting gravimetric datasets (Kissling and Schwendener, 1990). In this context, Schwenk et al. (2022a), Schaller et al. (2023) and Schuster et al. (2024) measured $\gamma$−density values on drill cores with a multi-sensor core logger. Their results revealed a strong dependence of the material densities on lithology, with the largest density values measured for gravel layers. Yet in addition to lithological control, Schwenk et al. (2022a) showed that the measured density values generally increase with the depth at which the Quaternary sequences were deposited, indicating that post-depositional compaction also played a role in determining the density of the Quaternary sediments (Schwenk et al., 2022a). However, for interpreting the gravity signal of Quaternary sediments, the bulk density of the entire sedimentary suite is more diagnostic than the density values of individual sedimentary beds (Kissling and Schwendener, 1990). Such bulk density values were determined by Bandou et al. (2022) for the Molasse bedrock and the Quaternary sediments overlying the overdeepened troughs using the results of a Nettleton profile across the Belpberg mountain (that is underlain by Molasse bedrock, Figure 2c), and through 3D gravity modelling. Using this approach, these authors assigned a bulk density of 2500 kg/m$^3$ to the Molasse units (Figure 2c). This is a substantially higher value than the bulk densities between 2150 and 2000 kg/m$^3$, which have been determined for the basal part and the top sequences of the Quaternary suites in the Aare main overdeepening, respectively. In particular, Bandou et al. (2023) documented that the best-fit reproduction of the gravity signals along the Bremgarten, Bern1, Bern2 and Kehrsatz profiles could be achieved by assigning a density value of 2000 kg/m$^3$ to the topmost sediments postdating MIS 6, and a higher density of approximately 2150 kg/m$^3$ (due to a greater compaction) to the underlying Quaternary deposits predating MIS 6. Drilling information (Mattenhof, Marzili, Metas) shows that the sediments younger than MIS 6 comprise a suite made up of gravels (Mattenhof), alternations of gravel, mud and sand beds (Marzili) and mud with interbedded sand layers (Metas). The results thus indicate that the bulk densities of the Quaternary sediments depend less on the lithological architecture of the material or the depositional environment in which the sediments were deposited. Instead, they appear to be primarily influenced by the overburden of the overdeepening fill and the number of glaciations, during which the Quaternary sediments were compacted under a thick glacial cover (Bandou et al., 2023). For instance, a sequence postdating MIS 6 was compacted by a piedmont glacier during the Last Glacial Maximum (LGM) only, while the older sediments experienced a glacial compaction during at least two full glaciations.

*2.6 Riegels and slot canyons in Alpine valleys*

Bedrock swells between neighbouring basins are common features in previously glaciated landscapes (Anderson et al., 2006; Alley, 2019). They are common in the European Alps (see Figure 3, for a few examples), and they have also been detected underneath active glaciers (Feiger et al., 2018; Nishiyama

et al., 2019). In the Alps, most of the bedrock swells cross the thalweg of valleys (Figure 3) and are
dissected by inner gorges or slot canyons that connect the upstream with the downstream basin (Hantke
and Scheidegger, 1973; Valla et al., 2010; Montgomery and Korup, 2011). In addition, Alpine bedrock
riegels have a geometry where the upstream stoss side is flatter than the downstream lee side (insets of
Figure 3). This is particularly the case for the swells in (Figure 3): the Aare valley (Figure 3a; dip of
stoss side and lee sides <5° and >6°, respectively; Hantke and Scheidegger, 1973), the Trift valley
(Figure 3b; c. 30° versus 40°; Steinemann et al., 2021), and the Maggia valley (Figure 3c; 6° versus
40°). Bedrock riegels and slot canyons are also found on the foreland plateau adjacent to the Alps such
as the example east of Lucerne (Figure 3d), yet their geometric expressions are less well-developed. In
this work, we will document that the overdeepening beneath the city of Bern shares the same geometric
properties as the ensemble of bedrock riegels and slot canyons in Alpine valleys.

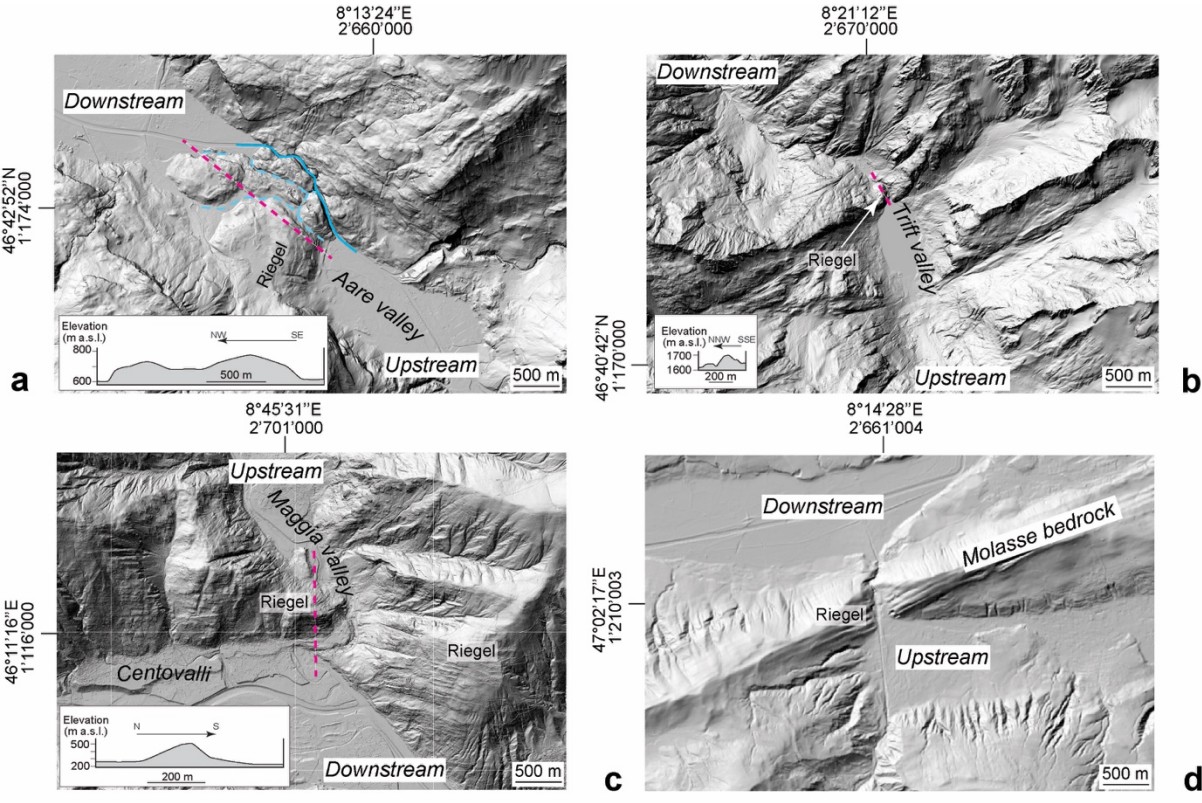

Figure 3

Figure 3:   Hillshade 2 m-SwissAlti3D DEM (© swisstopo) illustrating examples in Alpine valleys where
bedrock riegels separate overdeepened basins situated farther upstream and downstream. The
insets illustrate topographic sections across the riegels, and the arrows display the flow direction
of the glaciers during a glaciation. The coordinates refer to the Swiss coordinate system
(CH1903+). Longitudes and latitudes are also indicated.


**3      Dataset and Methods**
The bedrock topography beneath the city area of Bern was already reconstructed in 2010 and then
updated in 2016, based on information from thousands of drillings publicly available on the Geoportal
of the Canton Bern (see Dürst Stucki et al. (2010), and Reber and Schlunegger (2016), respectively).
Since these drillings primarily penetrated the entire Quaternary sequence down to the bedrock at the
lateral margins of the Aare main overdeepening, we consider the bedrock topography model of Reber
and Schlunegger (2016) for the shallow parts of the trough as accurate. Yet detailed reconstructions of
the deeper, central part of the overdeepening were hindered due to a lack of information from deep
drillings at that time (Reber and Schlunegger, 2016). Here, we benefit from the results of a recent gravity
survey conducted in the city area of Bern (Bandou et al., 2022; 2023; Bandou, 2023a) and information
from new drillings >50 m deep. We proceeded through compiling, as a first step, the publicly available
gravity data. We re-processed them to provide information about the spatial pattern of the gravity signal,
either from the bedrock topography beneath the overdeepening fill (section 3.1) or from the
overdeepening fill itself (section 3.2). Using these data along with the results from modelling conducted
by Bandou et al. (2023), we reconstructed a map outlining the general thickness distribution of the
Quaternary sediments (section 3.3). This was then used as the basis to update the existing bedrock
topography model of Reber and Schlunegger (2016), thereby incorporating data from >100 drillings
that penetrated >50 m into the subsurface (section 3.4).

*3.1      Assessing the gravity signal of the bedrock topography beneath the overdeepening*
We compiled the gravity data collected by Bandou (2023a) and combined them with data archived in
the Gravimetric Atlas of Switzerland by Swisstopo (Olivier et al., 2008; 2011). From this dataset, we
calculated the Bouguer anomaly values (see Bandou et al., 2023, for references to the methodological
papers) using the density of the Molasse bedrock (2500 kg/m$^3$) instead of the standard density of 2670
kg/m$^3$ that is conventionally used for Bouguer anomaly corrections. We then draw the isogals (contour
lines of equal Bouguer anomaly values) using the Golden Software Surface licensed to Swisstopo. This
map was used to infer the general shape of the bedrock topography beneath the overdeepening fill. In
particular, deviations of the isogals from the long-wavelength trend can serve as *a-priori* constraints for
reconstructing the course and geometry of the bedrock outlining the overdeepening.

*3.2      Assessing the gravity signal of the Quaternary sediments overlying the overdeepening*
Subtracting the Bouguer anomalies values measured along a profile from the regional gravity field along
the same profile yields what is referred to as the residual gravity anomaly. The related values provide
information about a near-surface body or structure with a bulk density different from that of the
surrounding bedrock (Kissling and Schwendener, 1990). Bandou et al. (2022; 2023) used this concept
to determine the gravity signal of the Quaternary sediments overlying the Molasse bedrock. They
proceeded by calculating the residual gravity anomaly values along 10 profiles perpendicular to the
inferred course of the Aare main overdeepening (black lines in Figure 2c). Note that because the
Quaternary deposits have a lower bulk density than the Molasse bedrock, the occurrence of such
deposits results in a negative residual gravity anomaly (Kissling and Schwendener, 1990). Accordingly,

a larger bulk mass of Quaternary sediments yields a stronger (and thus a more negative residual anomaly) signal than a fill with less Quaternary material (Kissling and Schwendener, 1990; Bandou et al., 2022). Following this concept, we compiled the residual anomaly data from Bandou et al. (2023) for each gravity profile and drafted a contour map where each line displays the same residual anomaly value. This map was drawn by hand, thereby considering the *a-priori* information about the orientation of the Aare main overdeepening (Reber and Schlunegger, 2016).

*3.3    Estimating the thickness of Quaternary sediments based on gravity data*

Residual gravity anomaly values can be converted to thicknesses of Quaternary sediments through modelling, provided that *a-priori* data is available (Kissling and Schwendener, 1990). This includes information on: (i) density contrasts between the Molasse bedrock and the Quaternary fill, (ii) depths of bedrock encountered in drillings, and (iii) an already existing bedrock topography model (in our case the bedrock topography model of Reber and Schlunegger, 2016). Bandou et al. (2023) used a 3D gravity software referred to as PRISMA (Bandou, 2023b) to implement this approach, modelling the residual gravity anomalies along six profiles (Figure 5b) where the aforementioned *a-priori* data is well constrained. Note that upon using PRISMA, the geometry of the overdeepening fill was approximated by Bandou et al. (1922, 1923) through multiple right-handed prisms oriented as perpendicularly as possible to the profile of interest (Nagy, 1966; Banerjee and Das Gupta, 1977). We compiled the results of the PRISMA modelling presented by Bandou et al. (2022, 2023) to draw a map displaying the thickness distribution of Quaternary sediments overlying the Aare main overdeepening. When creating this map, we considered that a trend towards smaller or larger negative residual anomalies indicates a thinning or thickening of the Quaternary sediments, respectively (Kissling and Schwendener, 1990; Bandou et al., 2023). The difference between the elevation of the modern topography and the thickness of the Quaternary sediments returns a map displaying the bedrock topography.

*3.4    Combining the results of the gravity survey with drilling data to reconstruct the details of the bedrock topography*

We updated the bedrock model of Reber and Schlunegger (2016) with information about the general shape of the overdeepening retrieved through gravity modelling outlined above, and we additionally considered the information of >100 drillings that were sunk >50 m deeply into the subsurface during the past years. Similar to Reber and Schlunegger (2016), we manually drew the isohypses of the bedrock, inferring that changes in the orientation of the contour lines and the depths of the bedrock were gradual. We finally combined the map displaying the geometry of the bedrock beneath the overdeepening with the elevation data provided by the 2 m-SwissAlti3D DEM (based on LIDAR data of Swisstopo) to present the shape of the bedrock topography as shaded relief.

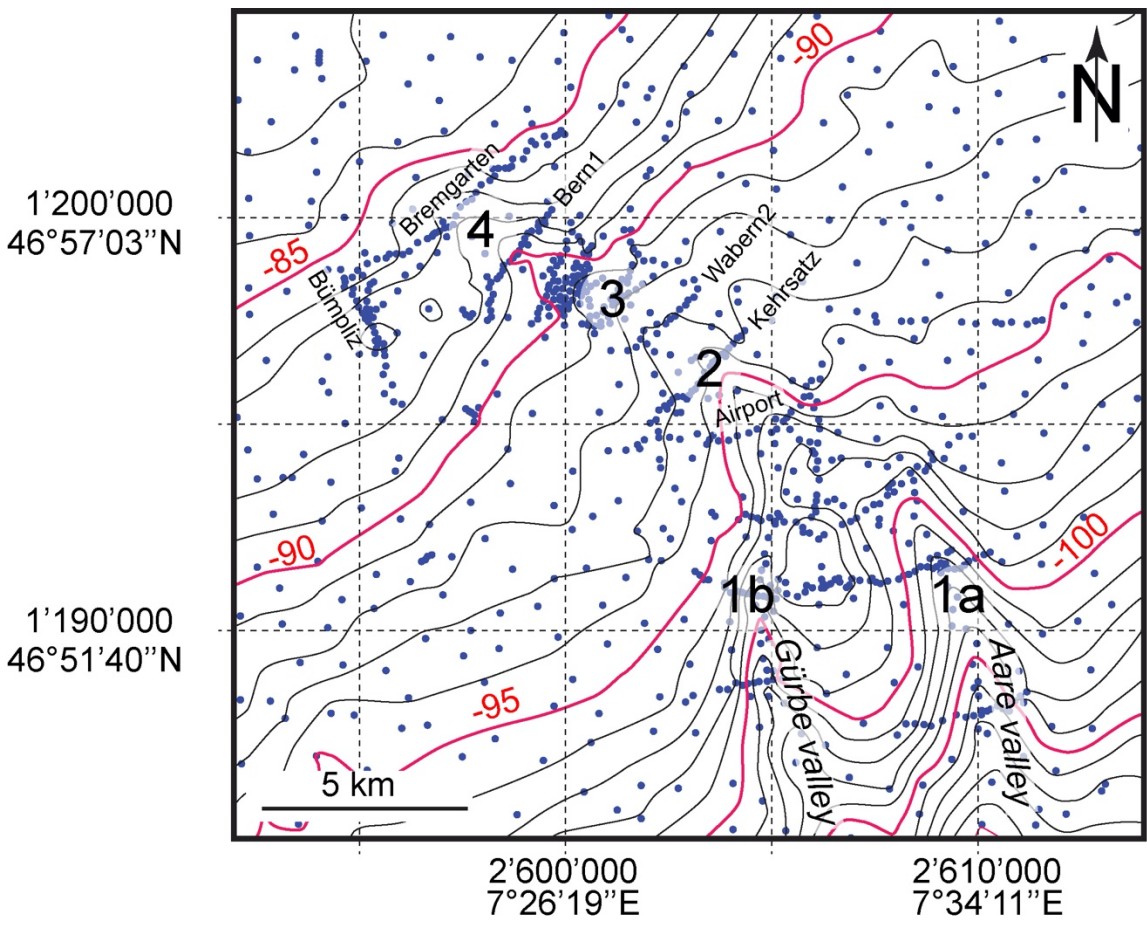


Figure 4: Bouguer anomalies, calculated with the density of the Molasse bedrock (2500 kg/m³). The blue dots are gravity data taken from the Gravimetric Atlas of Switzerland (Olivier et al., 2008; 2011; Swisstopo) and from Bandou (2023a). The isogals, indicated in mGal, illustrate the general gravity trend in the region and deviations thereof. 1a and 1b are sites located in the Aare and Gürbe valleys, respectively. These are the locations where the isogals have the largest deflections from the large-wavelength trend. Farther to the N (site 2) and then to the NW, the deflections decrease, reaching the lowest values at site 3. They increase again towards site 4 and then fade towards the NW. The figure also shows the locations of the gravity profiles presented in Bandou (2022) and Bandou et al. (2023). The grid corresponds to the Swiss coordinate system (CH1903+). Longitudes and latitudes are also indicated.


**4     Results**
*4.1     Isogals and gravity signal of the bedrock topography beneath the overdeepening*
The isogals calculated with the density of bedrock (2500 kg/m³) clearly depict the general gravity trend,
which is characterized by a continuous SE-directed increase of the Bouguer anomaly values from -85
mGal in the NW to -105 mGal towards the SE (Figure 4). Note that a more negative value implies a
stronger gravity anomaly. The isogals generally strike SW-NE, reflecting the orientation of European
continental lithosphere, which gently dips beneath the Alpine orogen. However, and most importantly
in our context, the isogals also deviate from this pattern by being deflected towards the NW, where we
expect the occurrence of the Gürbe tributary trough and the Aare main overdeepening. This anomaly
(or deflection) has indeed the largest amplitudes of >4 mGal and >3 mGal beneath the Aare (location
1a on Figure 4) and Gürbe valleys, respectively (location 1b). This indicates that the depth of the
overdeepened trough is greatest there. Farther to the NW, the amplitude of the deflection decreases from
approximately 3 mGal at site 2 (between Airport and Kehrsatz) to <1 mGal at site 3 (Figure 4),
suggesting a shallowing of the bedrock trough and thus the occurrence of a swell (or riegel). From there,
the amplitude increases again at site 4 as the trough appears to deepen once more, after which the
anomaly fades farther to the NW.

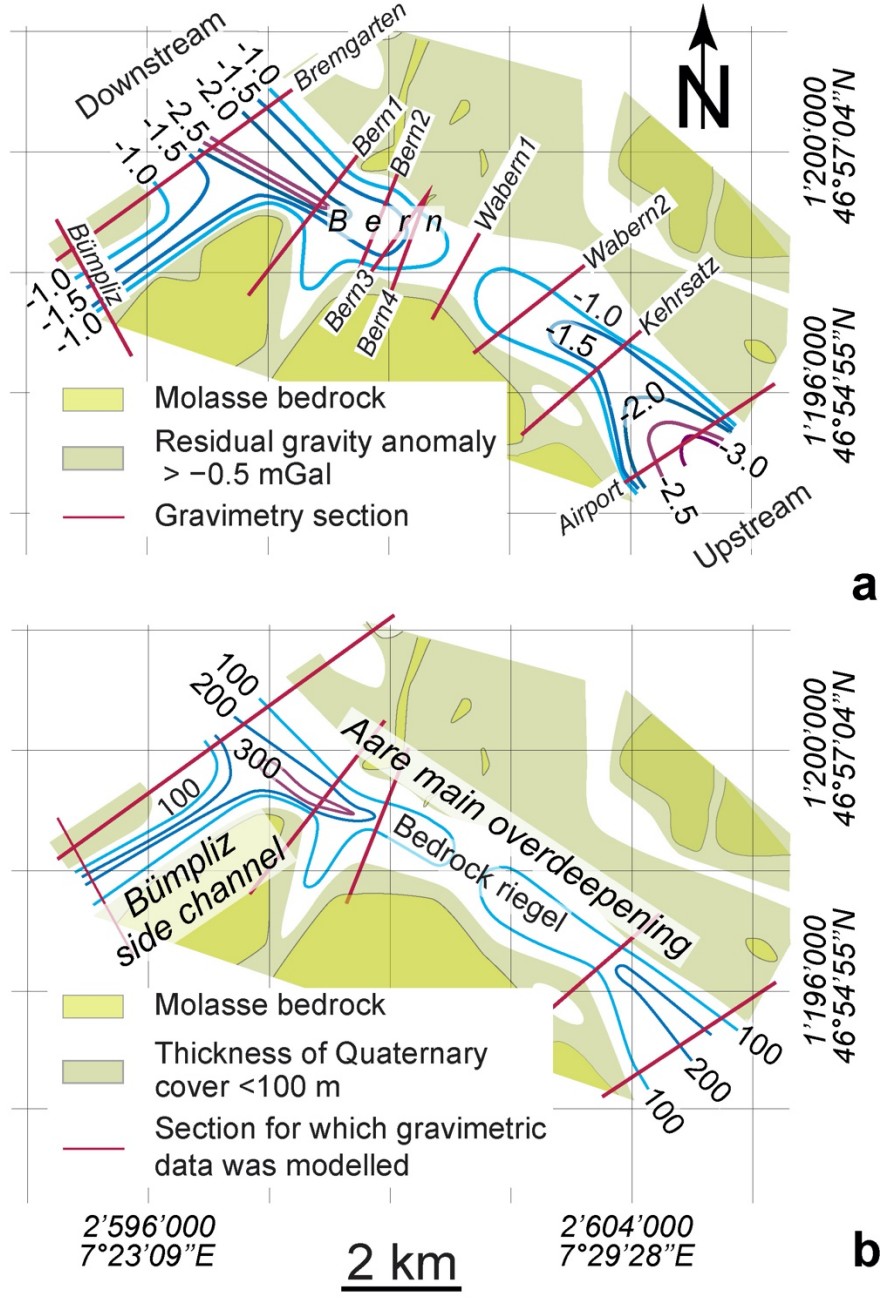

Figure 5


Figure 5: Residual gravity anomalies, representing the gravity signal of Quaternary sediments, and inferred thicknesses of Quaternary deposits. a) The contour lines of the residual gravity signals (mGal) caused by the Quaternary fill of the Aare main overdeepening are mainly based on gravity surveys along 10 sections (red lines; Bandou et al., 2023). Here, more negative values imply a greater signal caused by the bulk mass of Quaternary sediments overlying the overdeepened trough (Kissling and Schwendener, 1990; Bandou et al., 2022). b) Spatial distribution of Quaternary sediments, here expressed by the related thicknesses. These are mainly based on the results of gravity modelling, where Quaternary mass and its spatial distribution was forward modelled until a best-fit between the modelled and observed gravity signals of the Quaternary mass overlying the overdeepened trough was reached (Bandou, 2023; Bandou et al., 2023). Note that only the residual gravity anomalies of the Airport, Kehrsatz, Bern2, Bern1, Bremgarten and Bümpliz sections were modelled by Bandou et al. (2023). The grid refers to the Swiss coordinate system (CH1903+). Longitudes and latitudes are also indicated.


*4.2 Gravity signals of the Quaternary sediments overlying the overdeepening*
The residual gravity anomalies, which correspond to the gravity signal of the Quaternary sediments,
reveal the same pattern as the isogals where the Bouguer anomaly values were calculated with the
bedrock density of 2500 kg/m³. For the section across the Gürbe and Aare valleys (Figure 2c), Bandou
et al. (2022; 2023) showed that the Quaternary fill of the Aare main overdeepening results in a gravity
signal that ranges between c. -4.0 and -0.5 mGal. In addition, they showed that this signal changes from
upstream to downstream: In particular, along the Gürbe-Aare transect (Figure 2c), which also crosses
the Belpberg mountain ridge made up of Molasse bedrock, the strength of the signal ranges from c. -
2.9 mGal in the Gürbe valley to c. -4.1 mGal in the Aare valley (Bandou et al., 2022). Farther
downstream, the residual anomaly values and thus the signal of the overdeepening fill decreases, where
the corresponding values change from c. -3.0 mGal (Airport profile) to approximately -1.5 and finally
c. -1.0 mGal along the Kehrsatz and Wabern2 profiles, respectively (Figure 5a). The weakest signals
with values between c. -0.5 mGal and -1 mGal were reported for the Wabern1 profile (Bandou et al.,
2023; Figure 5a). This suggests a decrease in the mass of Quaternary sediments approaching Wabern1,
most likely due a shallowing of the bedrock forming a riegel in this area. Farther downstream, the
gravity signal of the Quaternary fill increases again and reaches values between c. -1.0 and c. -2.0 mGal
along the Bern sections, and then approximately -2.5 mGal along the Bremgarten section c. 2 km farther
downstream. This points towards an increase in the Quaternary mass and thus towards a deepening of
the trough in this direction. The residual anomaly data thus clearly depict the course of the Aare main
overdeepening, which strikes SE-NW in the city area of Bern (Figures 2c, 5a). Towards the NW margin
of the study area, a second overdeepening referred to as the Bümpliz tributary trough (Schwenk et al.,
2022a) strikes SW-NE and converges with the Aare main overdeepening NW of Bern. The gravity
signal of the Bümpliz sedimentary fill is less and reaches a value of c. -1.5 mGal (Figure 5a; Bandou et
al., 2023). Finally, the upstream side of the inferred bedrock riegel dips gentler than the downstream
side, which is twice as steep: on the stoss side, the residual gravity anomalies change from <-2.5 mGal
to >-1.0 mGal over a downstream distance of c. 4 km whereas on the lee side, the same change in the
gravity signal occurs over only 2 km. Given that the residual gravity signal is a direct response of the
bulk mass of Quaternary sediments overlying the Molasse bedrock (see section 4.2), and thus their
volume supposing a lower density than the Molasse bedrock (see next section and Bandou et al., 2022;
2023), the differences in the upstream and downstream gradients of the residual gravity anomaly values
disclose the contrasts in the dip angles of the bedrock topography.

*4.2      Thickness of Quaternary sediments*
Available drilling information shows that the Quaternary fill in the Bern region generally consists of an
alternation of gravel, sand and mud (Figure 2d), which have a bulk density that ranges from 2150 kg/m$^3$
for material at the base of the overdeepening fill, to 2000 kg/m$^3$ for the sediments towards the top
(Bandou et al., 2023). Based on a sensitivity analysis where the gravity response to different densities
for the Quaternary sediments was evaluated, Bandou (2023a) and Bandou et al. (2022, 2023) could
exclude the possibility that the Bouguer anomaly and residual anomaly patterns displayed in Figures 4
and 5a could be explained by spatial differences in the sedimentary architecture of the Quaternary fill.
For instance, the low residual gravity anomalies displayed in the region of the Wabern2 profile (Figure
5a) would require an amalgamation of highly compacted glacial till. However, this is not consistent
with the stratigraphic log of the core drilled at Metas (Figure 2d), which is made up of an alternation of
sand, mud and gravel that was most likely deposited in a lacustrine environment. Instead, we prefer a
perspective where the pattern of residual gravity anomaly values reflects spatial variations in the
thickness of the Quaternary sediments. Accordingly, the thickest Quaternary suite can be found
upstream and downstream of Bern (Figure 5b), where the Aare main overdeepening is between 4 and 5
km wide and >200 m deep, consistent with drilling information (Bandou et al., 2023). In the city area
of Bern, however, the main trough tends to become shallower. This is indicated by the thickness of the
Quaternary sediments reaching 100 m and possibly less (Figure 5b). The thickness of the Quaternary
sediments filling the trough then increases again farther downstream.

*4.3      The consideration of deep drillings discloses the occurrence of slot canyons*
The reconstructed bedrock topography of the target region reveals a complex pattern (Figure 6), which
can be described as a bedrock riegel that is dissected by multiple, partly anastomosing slot canyons or
inner gorges (Bandou et al., 2023). At this stage, we cannot precisely reconstruct the number of the
inferred canyons because we lack a high-resolution database of deep drillings (Figure 6). Yet, the
discrepancy between (i) a relatively low gravity signal particularly between the Wabern2 and the Bern
sections (Figure 5a) and (ii) drillings that reached the bedrock at much deeper levels >200 m below the
surface (Figures 6) can only be resolved by invoking the occurrence of a plateau at shallow elevations
that is dissected by one or multiple slot canyons (Figure 7). These gorges are up to 150 m deep and may
connect the overdeepened basins upstream and downstream of the city area of Bern. In particular, south
of Bern along the Aare profile (Figures 2b and 8a), the Aare main overdeepening has a cross-section
that displays two superimposed levels of U-shapes, each of which with steep lateral flanks and a flat
base. While the upper flat base occurs at an elevation of c. 450 m a.s.l., the lower flat contact to the
bedrock is situated at c. 250 m a.s.l. and thus approximately 200 m deeper than the upper level (Bandou
et al., 2022). Approximately 5 km farther downstream along the Airport section (Figures 2b, 8b), the
cross-sectional geometry of the Aare main overdeepening maintains its generally U-shaped geometry
with a base at an elevation between 200 and 250 m a.s.l. There, the base of the overdeepening appears
less flat than farther upstream, but we acknowledge that the density of deep drillings in the region
(Figure 6) and the resolution of the gravity data (Figure 5a, Bandou et al., 2023) is not high enough to
fully support this comparison. Upon approaching the city area of Bern, the base of the bedrock becomes
shallower and appears to evolve towards a plateau particularly between the Kehrsatz and Bern2 sections
(Figures 6, 7, 8c, d and e). This plateau is situated at an elevation of c. 400 m a.s.l. (dashed lines on
Figure 8) and dissected by multiple slot-canyons, as evidenced by drillings reaching depths down to c.
300 m a.s.l. and even lower elevations, yet the canyons remain undetected by the gravity survey. This
implies that the canyons must be cutting up to 150 m deep below the plateau at c. 400 m a.s.l. and that
they are too narrow to be detected by the gravity survey (Bandou al., 2023). Farther to the Northwest
reaching the terminal part of the Aare main overdeepening (Figure 2b), the trough widens again and
gives way to a relatively deep basin where the deepest part occurs at an elevation of almost 300 m a.s.l.
(Figures 6, 8f). This terminal basin appears to be connected with the Bümpliz tributary trough farther
to the SW. Yet the density of drillings is too low (Figure 6) to determine whether a possible bedrock
swell separates the Aare main overdeepening from the Bümpliz tributary trough (Figure 2b).

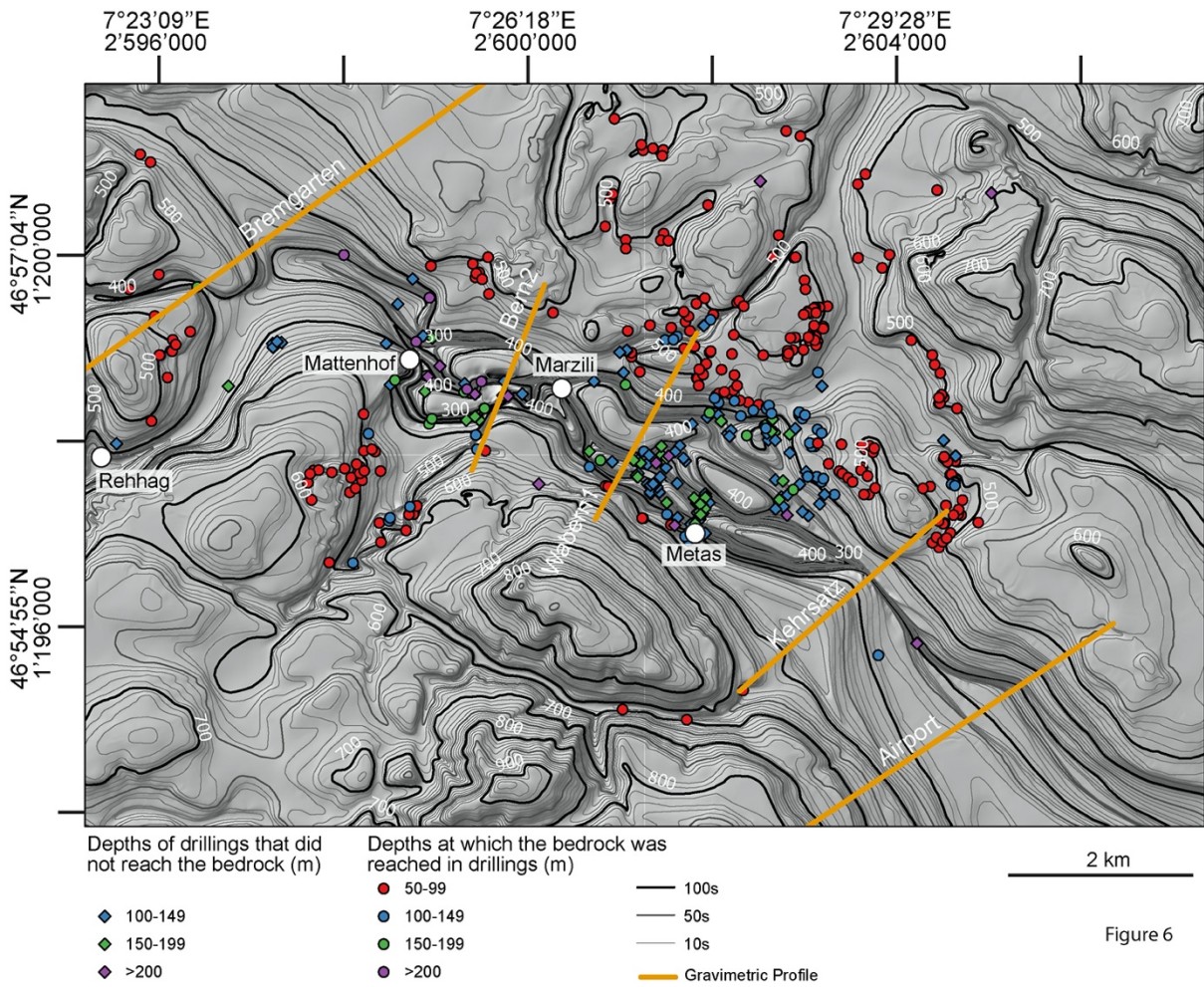


Figure 6:    Hillshade DEM, illustrating the bedrock topography of the Bern area, together with deep drillings that either reached the bedrock (circles) or that ended in Quaternary sediments (diamonds). The shallow drillings (<50 m) are not displayed on this map since the number is too large (more than 1000, please see Reber and Schlunegger, 2016). The isohypses were drawn for every 10 m. The coordinates along the figure margin refer to the Swiss coordinate system (CH1903+). The sections shown on this map are used to illustrate the cross-sectional geometry of the overdeepening beneath Bern (see next figures). The white circles represent those drillings, the logs of which are illustrated in Figure 2d.


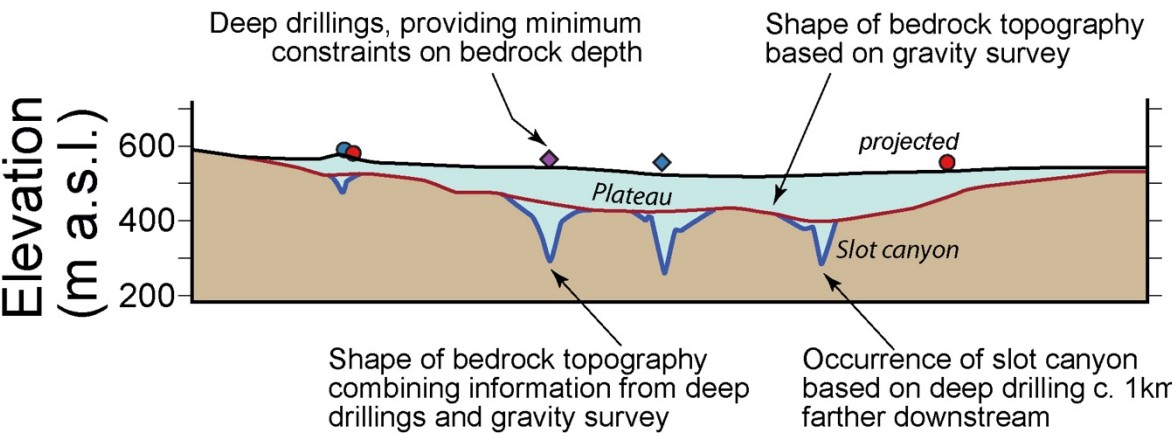

Figure 7:    Example that illustrates how we proceeded upon reconstructing the bedrock topography beneath
             Bern. We started with the general shape of the bedrock topography using the gravity signal of the
             bulk Quaternary mass as a basis (red line, and Figure 5b). Information from drillings >50 m deep
             (circles and diamonds: see Figure 6 for explanation of colors) allowed then to reconstruct the
             course and geometry of the slot canyons (blue line). The mass of their Quaternary fill is too low
             to be identified by the gravity survey. This is the case because the strength of a gravity signal
             decays exponentially with depth (see also Bandou et al., 2023, for further explanations).

## 5    Discussion

### 5.1    Limitations upon reconstructing the bedrock topography model

The inferred existence of a bedrock riegel and slot canyons below Bern is based on two features: (i)
gravimetric data showing a relatively low negative anomaly, which we interpret as a low depth to
bedrock in the Bern city area, and (ii) previous borehole logs that show a much greater drilled depth to
bedrock. Indeed, the combination of deep bedrock detected from borehole data in an area of otherwise
characterized by shallow bedrock, as imaged by gravitmetry, suggests that the canyons must extend
deeply while remaining highly confined in order to stay below the spatial resolution of the gravimetry
method. However, we acknowledge that no direct drilling evidence confirms the presence of such a
riegel. Nevertheless, the contour lines of the Bouguer anomaly values, calculated using a density of
bedrock (2500 kg/m$^3$), indicate that the target overdeepening is generally broad and deep upstream of
Bern, shallow beneath the city, and then narrows and deepens downstream of it (Figure 4). In addition,
gravity data collected at 10 gravity stations along the Bern2 profile does point towards the occurrence
of a residual anomaly signal with a short wavelength beneath the main large-wavelength residual gravity
anomaly (Figures S1a and S1b in the Supplement). Indeed, using the results of 3D gravity modelling,
Bandou et al. (2023) considered the large-wavelength anomaly to be the gravity response of the
Quaternary fill overlying the bedrock riegel, whereas the short-wavelength anomaly beneath it suggests
the possible presence of a slot canyon filled by Quaternary sediments (Figure S1c in the Supplement).

Further slot canyons could not be identified upon modelling due to a lack of resolution of the gravimetric

data.

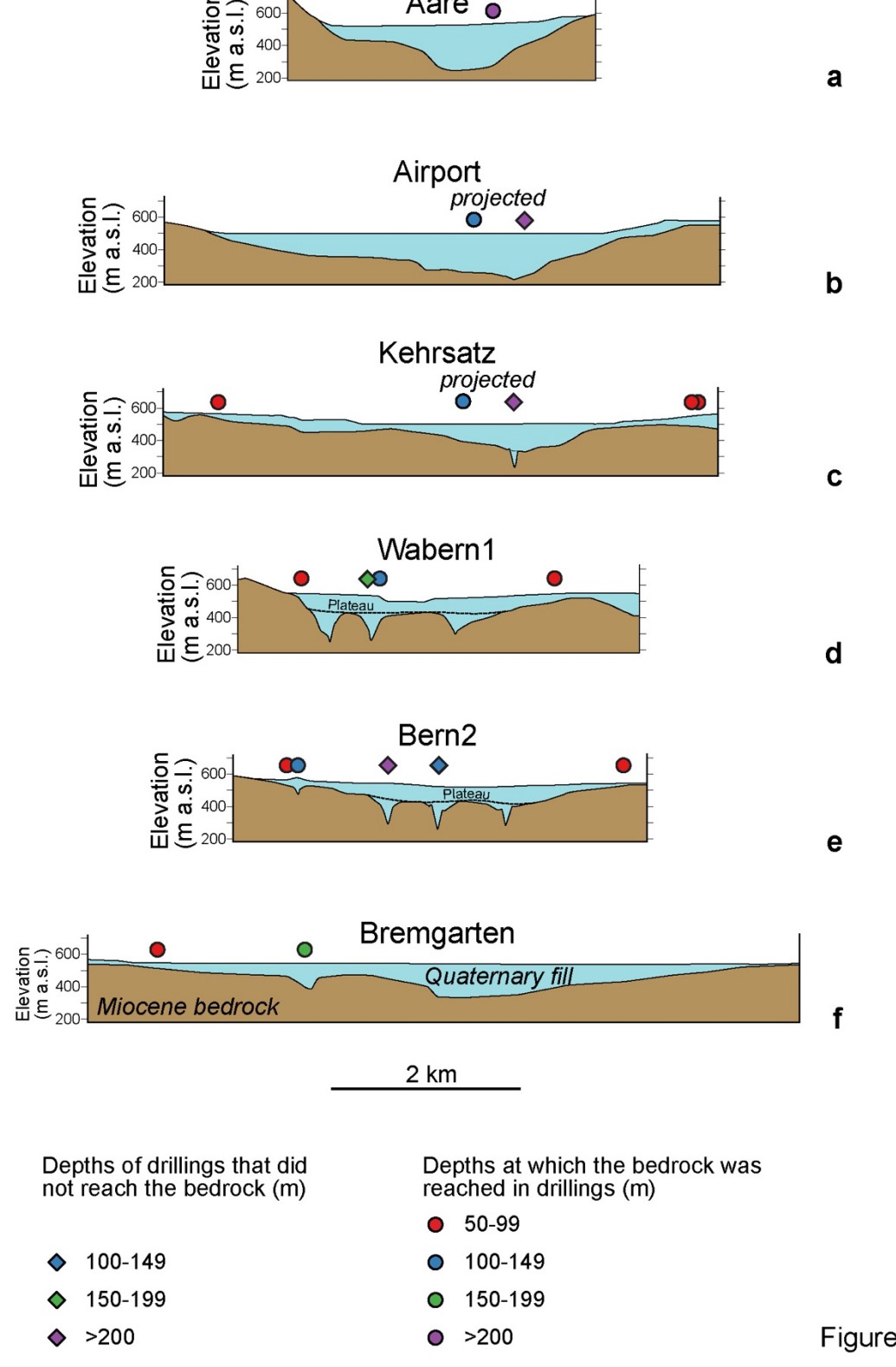

Figure 8: Sections through the Bern area, where the geometry of the bedrock is taken from the DEM illustrated in Figure 6. The Aare section is taken from Bandou et al. (2022). See Figures 2 and 6 for location and orientation of sections.

In summary, we are confronted with the situation that there is most likely a bedrock riegel imaged by the gravity data, and that thick Quaternary deposits (deep erosion) have been encountered in some deep drillings as well (and have also been detected in the Bern2 gravity profile; Figures S1a to S1b in the Supplement). We thus propose an interpretation where a bedrock riegel is cut by narrow slot canyons filled with Quaternary sediments, as such a scenario adequately combines the findings from our gravity survey and drilling information. Furthermore, using the modern examples such as the Aare gorge displayed on Figure 3a as a basis, we interpret that these slot canyons formed the hydrological link between the upstream and downstream basins. We exclude an alternative interpretation where the drilled Quaternary sequences represent the filling of isolated glacial potholes. Indeed, the short distance between the individual boreholes with thick Quaternary sequences and the almost linear arrangement of these boreholes, particularly near Wabern1 (Figure 6), suggests that the drilled sequences comprise the fill of continuous channels rather than potholes.

*5.2    Subglacial origin and the role of subglacial meltwater*

It is agreed upon in the literature that the formation of overdeepened basins can be understood as the response of erosion by glaciers. The main arguments that have been put forward are (i) the depths of the base of these depressions, which are generally below the current fluvial base-level, and (ii) the occurrence of adverse slopes in the downstream direction of these basins (Figure 1, Preusser et al., 2010; Patton et al., 2016; Alley et al., 2019; Magrani et al., 2022; Gegg and Preusser, 2023). Such geometric features are also encountered for the Aare main overdeepening beneath the city area of Bern. Therefore, the origin of this depression has repeatedly been interpreted as the response of the erosional processes of a glacier with a source in the Central Alps of Switzerland (Dürst Stucki et al., 2010; Preusser et al., 2010; Reber and Schlunegger, 2016; Magrani et al., 2022; Bandou et al., 2023). As a refinement already outlined by Bandou et al. (2023) and further detailed in this work, the overdeepening beneath Bern can be subdivided into a southeastern and a northwestern sub-basin. These depressions are separated by a bedrock riegel or swell, which itself is dissected by one or multiple slot canyons establishing a hydrological link between the upstream and downstream basins (Figure 6). Such ensembles of basins, riegels and slot canyons (or inner gorges) are common features in Alpine valleys (Figure 3) and have therefore been the target of previous research. In this context, it was proposed that such gorges and riegels in the Alps were likely shaped during several glacial/interglacial periods (Montgomery and Korup, 2011), and that the incision of the canyons occurred during the decay of glaciers and ice caps, when large volumes of meltwater were released (Steinemann et al., 2021). As further, yet only partly related examples, erosion by subglacial meltwater was put forward to explain the formation of inner gorges at the margin of the Fennoscandian ice sheet (based on the pattern of surface exposure ages; Jansen et al., 2014), and such a mechanism was used to explain (i) the origin of the deep channels on the floor of the eastern English Channel, and (ii) the breaching of the bedrock swell at the Dover strait during the aftermath of the Marine Isotope Stage (MIS) 12 or a later glaciation (Gupta et al., 2007;

Cohen et al., 2014; Gupta et al., 2017). In this context, Jansen et al. (2014) noted that typical field evidence for inferring a subglacial meltwater control includes (i) the occurrence of anastomosing channels, (ii) undulating valley long profiles, and (iii) a topography that apparently amplifies the hydraulic potential. The resolution of our data is not enough to see such details of the valley long profiles, but sufficient to display the anastomosing patterns of the slot canyons, with channels meandering, splitting and merging again (Figure 6).

*5.3     Formation through erosion by subglacial meltwater inferred from theory and modelling*

Besides the geometrical arguments and field-based observations outlined in the previous section, a subglacial meltwater influence on the formation of overdeepenings has also been inferred based on theoretical considerations, including the relationships between meltwater runoff and the sediment transport capacity of proglacial and subglacial streams (e.g., Boulton and Hindmarsh, 1987; Alley et al., 1997; Herman et al., 2011, Beaud et al., 2016). Because sediment transport increases exponentially with both the volume and seasonality of meltwater runoff, Alley et al. (1997) interpreted that subglacial and proglacial streams are among the most efficient sediment-transport mechanisms on Earth. This process peaks in the ablation zone of a glacier, where surface melt reaches the bed and significantly contributes to the generation of subglacial runoff. Also on theoretical grounds, Cohen et al. (2023) showed that subglacial meltwater is able to remove the sediment from the base of a glacier and to further incise into bedrock provided that the pressure of the subglacial meltwater and that of the ice overburden are at least the same (Boulton and Hindmarsh, 1987). The results from the model of Cohen et al. (2023), tailored to determine the location of the subglacial drainage pathways, further suggest that such conditions most likely prevailed at the front of piedmont glaciers and particularly during the decaying stage of a glacier when large volumes of meltwater were available. In addition, the model predicts that under such circumstances, the locations of subglacial meltwater pathways are likely to coincide with segments where high rates of glacial erosion occur (Cohen et al., 2023). Therefore, reaches with evidence for intense erosion by both water and ice occur in the same area and are hydrologically connected with each other. We propose this to be the case for the ensemble of overdeepened basins and slot canyons beneath Bern.

*5.4     The role of bedrock strength and the confluence of two glaciers*

The formation of riegels and basins is consensually understood as conditioned by differences in bedrock strengths. This also concerns the controls on the size of a basin itself where bedrock with a high erodibility tends to host a larger basin than lithologies where the erodibility is low (e.g., Magrani et al., 2020; Gegg and Preusser, 2023). Following this logic, swells preferentially would form in locations where the bedrock has a lower erodibility than the rock units farther upstream and downstream. This has been documented for the riegel in the Aare valley, which separates an overdeepened basin upstream

from a wide valley farther downstream (Figure 3a). There, the bedrock riegel is made up of the Quinten Formation (Gisler et al., 2020; Stäger et al., 2020). These limestones tend to have a lower erodibility (Kühni and Pfiffner, 2001) than the sandstone-marl alternations (North Helvetic Flysch; Gisler et al., 2020; Stäger et al., 2020) downstream of the bedrock swell, and the suite of sandstones, marls and dolomite beds upstream of it (Mels- and Quarten Formations; Gisler et al., 2020; Stäger et al., 2020). Another example is offered by the riegel in the Trift valley (Figure 3b), where the bedrock forming the ridge is made up of a banded, biotite-rich gneiss (Erstfeld gneiss). Upstream and downstream of the swell, the bedrock is cut by multiple faults and fractures, thus offering a lower resistance to erosion (Steinemann et al., 2021). In the Bern area, the bedrock architecture is comparable to the examples explained above where the UMM, which has a low erodibility, forms the swell, whereas the LFM with a relatively large erodibility constitutes the bedrock downstream of the riegel (section 2.3). In addition, the NW-SE striking faults in the Molasse bedrock (Isenschmid, 2019), which offer zones of mechanical weaknesses, most likely controlled the course of the slot canyons as they have the same orientation. Presumably as important as the contrasts in bedrock erodibility: the bedrock swell underneath Bern is situated in the confluence area between the Valais and Aare glaciers (Figure 2b). The occurrence of swells at the confluence area is consistent with observations in Alpine valleys (Figure 3) and with topographic and bathymetric DEMs of overdeepenings in Labrador, Canada (Lloyd et al., 2023). In this case, the deep carving into the bedrock would be the result of an acceleration of the ice flow (Herman et al., 2015) in response to the increase in the ice flux downstream of the confluence region. Alternatively, a bedrock riegel could also form upstream of the confluence of two glaciers (see e.g., the Maggia valley as modern example, Figure 3c). For the Bern area, the damming of the Aare glacier by the much larger Valais glacier could have caused a reduction of the flow velocity of the Aare glacier (Figure 2b). Consequently, the shear velocity and thus the bedrock abrasion rates would decrease, thereby facilitating the preservation of a bedrock swell.

*5.5     Differences in the geometries between the exposed riegels and basins in Alpine valleys, and the overdeepening beneath Bern*

Despite obvious similarities between the geometric properties of the overdeepening system beneath Bern and the currently exposed riegels and slot canyons in Alpine valleys, there are also major differences (Figure 3 versus Figures 6 and 8). The most striking one is the occurrence, beneath Bern, of the riegel and inner gorges approximately 50-100 m below the current base-level, and the absence of an obvious continuation of the thalweg NW of Bern (Figure 2c). Accordingly, the inferred interpretation where the slot canyons beneath Bern were formed by subglacial meltwater requires a mechanism where the meltwater is not only capable to incise into bedrock beneath a glacier, but also to escape the depression by ascending nearly 200 m from the base of the overdeepening to the surface near the glacier's snout. Using Bernoulli's principle as a basis (e.g., Batchelor, 1967), it was proposed that such

an ascent of subglacial meltwater was driven by the translation of high hydrostatic pressure into
hydrodynamic pressures at the downstream margin of a glacier (Dürst Stucki and Schlunegger, 2013).
Such a mechanism is most effective at work where the surface slope of a glacier is steeper than the
adverse slope of an overdeepening (Hooke and Pohjola, 1994), as is commonly found in the frontal part
of a glacier (Figure 1a). Since the ratio between the densities of ice and water is >0.9 (Harvey, 2019),
the inferred 200 m-rise of the meltwater requires a minimum hydrostatic pressure corresponding to
>220 m-thick ice to allow an upward water flow. Such a scenario is plausible, as the Are glacier in the
Bern area was estimated to have reached several hundred meters in thickness during past glaciations
(Bini et al., 2009; Preusser et al., 2011; Figure 2b). If this hypothesis is valid, then the thickness of the
piedmont glaciers sets an uppermost limit to the depth at which overdeepenings can be carved into the
bedrock, since it represents the driver of overpressure required for the subglacial meltwater to ascend
to the surface from deeper levels.

**6       Conclusions**
Bedrock riegels separating upstream and downstream basins are common features in modern Alpine
valleys, and they have been documented from overdeepenings in the region of Bern. We propose that
these riegels occur as ensembles together with slot canyons that cut through these swells and establish
a hydrological link between the upstream and downstream basins. We suggest this based on our
reconstruction of the bedrock topography of the Aare main overdeepening in the Bern area, and we
propose that such ensembles of basins, riegels and slot canyons also occur in other Alpine
overdeepenings such as the Rhone, Rhine and Inn valleys (Figure 9). We further suggest that these slot
canyons were formed through incision by glacial meltwater during the deglaciation when large volumes
of meltwater were available. As the flow must counteract adverse slopes, it may also be envisioned that
the slot canyons formed during glacial maxima, when ice thickness (and thus excess hydrostatic
pressure) is maximum, driving vigorous underflows. For the bedrock swell underneath Bern, the
resolution of the dataset presented in this work does not allow to locate and reconstruct the precise
course of the inferred slot canyons. However, the presented reconstruction of the bedrock topography
reconciles (i) the occurrence of low residual gravity anomalies in the Bern area (Figure 5a), which
suggests a topographic high of the incised bedrock marking the base of the overdeepening, and (ii) the
significant depth at which Quaternary sediments were encountered in drillings, indicating deep-reaching
bedrock incision (Figures 6, 7). In many Alpine valleys, such ensembles of riegel and slot canyons
appear to be preferentially formed in the confluence area between two glacial valleys and where the
bedrock has a relatively low erodibility. We posit that this configuration is also valid for the
overdeepening below the Bern area, where such a bedrock swell appears to be situated just upstream of
the confluence between the Aare and Valais glaciers, at least during LGM times and possibly during
previous glaciations. In addition, the inferred bedrock riegel beneath Bern is located where the bedrock
has a lower erodibility than farther downstream.

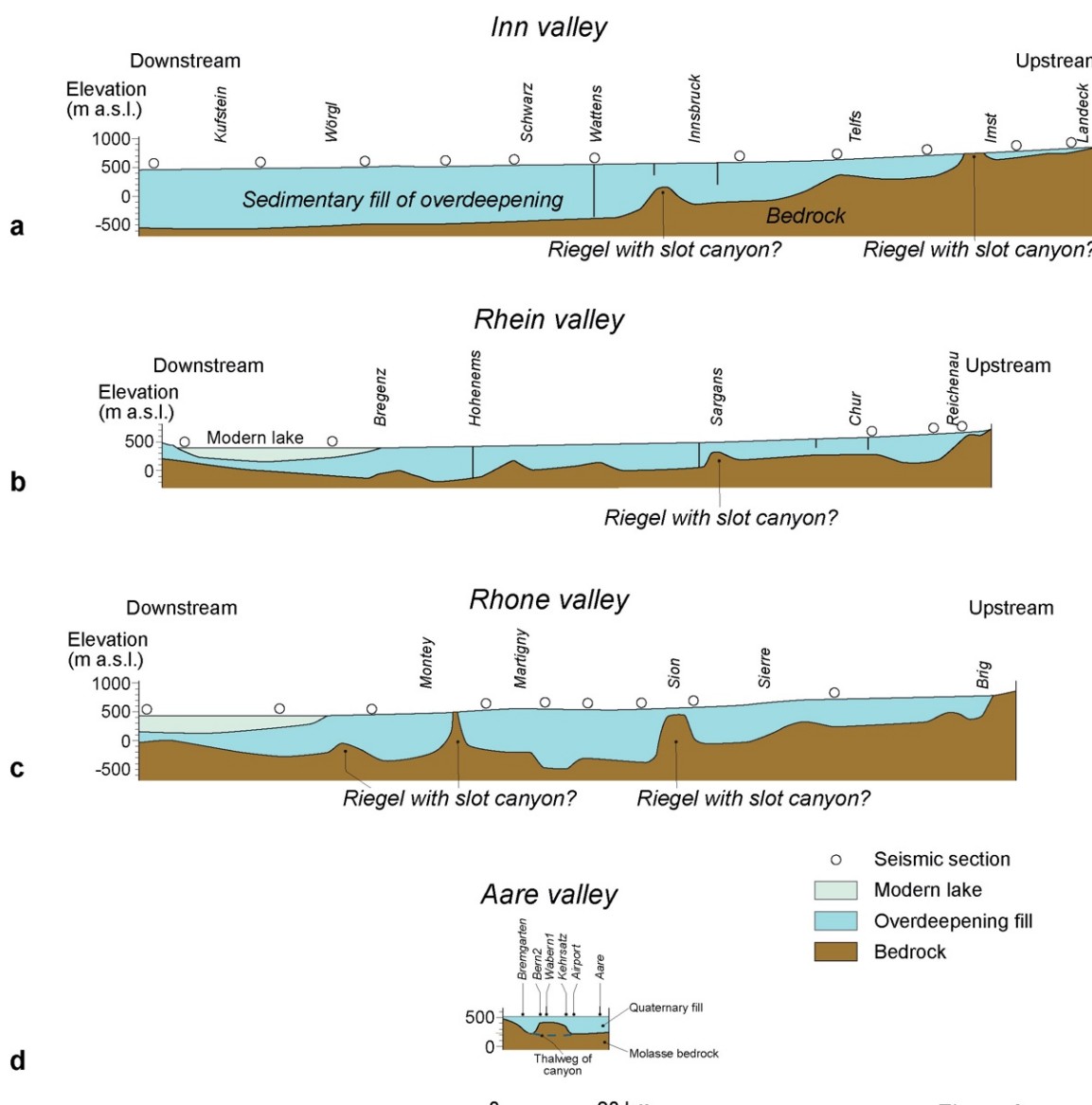


Figure 9:   Sections showing the patterns of overdeepenings from upstream to downstream for a) the Inn
            valley, b) the Rhine valley, c) the Rhone valley and d) the Aare valley in the Bern area. The
            examples of the Inn, the Rhine and the Rhone valleys are taken from Hinderer (2001), whereas
            the section along the Aare valley is a modified version of Bandou et al. (2023) and bases on the
            data presented in Figure 6. The data from the Aare valley covers a short distance only, but it shows
            a striking similarity to the riegels in the large Alpine valleys. Therefore, it is quite likely that the
            other riegels are also dissected by narrow channels and that all settings share a similar origin.


In summary, we present a bedrock model that documents an upstream-downstream trend of the
subglacial drainage network: (i) Along the Aare cross-section, which is situated upstream of the riegel
there appears to be no evidence of a channelised subglacial drainage network incising into the bedrock;
(ii) in the area of the inferred riegel, we postulate the occurrence of an anastomosing network of slot
canyons based on drilling information, which evolves (iii) downstream of the riegel into a single canyon
as seen along the Bremgarten cross-profile. This rises further questions about the mechanisms that could

be responsible for these changes in the network, how such processes evolved in space and time, and how possible variations in the subglacial drainage network would have affected bedrock erosion and ice flow. Answers to such following up questions require detailed constraints on the ages and the sedimentary architecture of the Quaternary fill, which are not available. Yet, the few chronological information published on the Quaternary fill of overdeepenings in the Swiss Plateau does support an interpretation where the deep carving occurred during multiple stages since the Middle Pleistocene Transition c. 800 ka ago (Schlüchter, 2004). Apparently, the change in the frequency of glacial-interglacial cycles from a 40 ka- to a 100 ka-periodicity, which occurred at that time, not only resulted in rapid glacial erosion (Pedersen and Egholm, 2013) and in the deep glacial carving of U-shaped valleys in the Alps (Häuselmann et al., 2007, Valla et al., 2011), but also in the formation of overdeepenings with complex geometries including basins, riegels and slot canyons in the foreland.

**Acknowledgement**

This work was financially supported by the Swiss National Science Foundation (project No. 200021_175555) with contributions from the Stiftung Landschaft und Kies, swisstopo and the Gebäudeversicherung Bern GVB.

**Data availability**

All data used in this paper can be ordered by the Authorities of the Canton Bern and by the authors on request.

**Autor contributions**

EK designed the study, together with FS and DB. DB collected the gravity data and processed them, with support by UM and EK. FS wrote the paper and conducted the analyses and interpretation of the data. RR drafted the bedrock topography map. PS, MS, DM and GD contributed to the discussion. All authors approved the article.

**Competing interests**

The authors declare that they have no conflict of interest.

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
