# Peer review of "The Aare main overdeepening on the northern margin of the European Alps: Basins, riegels,"

_EGUsphere, 2024_

## Referee Comment (RC2)

**Review of manuscript 'Overdeepening or tunnel valley of the Aare glacier on the northern margin of the European Alps: Basins, riegels, and slot canyons' by Schlunegger et al.**

The manuscript 'Overdeepening or tunnel valley of the Aare glacier on the northern margin of the European Alps: Basins, riegels, and slot canyons' by Schlunegger et al. reviews and synthesizes a combination of existing data to investigate the Aare overdeepening in the Bern area. Gravimetry data is used to reconstruct the thickness of Quaternary sediment infill and is used to identify two basins separated by a bedrock riegel. Drilling data indicates deeper than expected bedrock which is interpreted to represent a series of slot canyons cut into this bedrock riegel by subglacial meltwater during deglaciation. A comparison to other alpine valleys is then made to highlight how the location of bedrock riegels formation tends to be related to the confluence zones of alpine glaciers and the relative difference in erodibility of the bedrock. Additionally, the same bedrock riegel/slot canyon landform assemblage is suggested to occur in these other alpine valleys. While the findings are relevant to readers of Earth Surface Dynamics and worthy of publication, I believe that there are a series of issues with the manuscript that should be addressed prior to publication. I list these comments below.

**Major comments/questions:**

Reviewer 1 has already raised the issue of whether the interpreted riegel could be the result of either an underestimation of the Quaternary infill by gravimetry and the misinterpretation of drillings and I note that you have already responded to these comments. So I don't want to dwell on this topic too much. But I welcome the introduction of new data that would support your interpretation and while I don't necessarily disagree with your interpretation, I would appreciate a brief discussion of the possible alternative interpretations to highlight some of the uncertainties and be more transparent to the reader. So I look forward to reading this addition.

On Figure 7, I would appreciate the location of deep drillings being labelled (in a similar style to Figure 6). This would allow the reader to easily see the ground-truthed evidence supporting the relatively low Quaternary infill (if any boreholes do show this) and location of slot canyons. While this information can be found in Figure 5, it can be quite difficult to interpret this data from a map figure and I would appreciate it plotted onto the cross-profiles as well. For example, on the Kehrsatz cross-profile you reconstruct a single slot canyon but on Figure 5 there don't appear to be any drillings which would support this reconstruction? If there is a drilling that supports it then could you label it on Figure 7c, even if the drilling is one of those located a few hundred metres up or downstream of the drawn cross-profile.

Section 5.5 Chronological framework. A synthesis of dating evidence is mentioned in both the abstract and introduction of the paper, so I expected the discussion of the chronological constraints to be more important to the paper. But, this section feels a little bit like a brief review of the existing constraints and doesn't really significantly add anything new or contribute to your aim of exploring the origin of these overdeepenings. While it is important to understand the timing of glaciations that formed these overdeepenings, I believe that this section could be simplified and cut to about half it's current length. In its current length I felt like it distracted from the main message of the manuscript. In your conclusions section, you barely mention the chronology and I think this illustrates how this section isn't a key element of your manuscript and it's location in the discussion section is taking away from the message. Another alternative to shortening this section could be moving it earlier in the manuscript, perhaps after Section 2 on 'Riegels and slot canyons in the Alpine valleys…'

The evolution of the slot canyon network across the bedrock riegel. Across Figure 7, there appears to be an evolution of the subglacial drainage network that incised these slots canyons as it passes over the riegel. At the Aare cross-section, upstream of the riegel there appears to be no evidence of a channelised subglacial drainage network incising into the bedrock (distributed subglacial drainage network or just no drilling data which identified a channel?). As you pass over the riegel there is the anastomosing network of slot canyons that you reconstruct. Then downstream of the riegel on the Bremgarten cross-profile, there is a single slot canyon (possibly?). I am aware that these reconstructed networks are very dependent on drilling data which struck one of these canyons and so this reconstruction has quite large uncertainties, but, I would be interested to see a description of this evolution of the channel network. If you believe that your data is reliable enough to reconstruct it, if not feel free to ignore this. What are the processes that could drive these changes in network and how it relates to the changing bed slope? Additionally, how could these variations in the subglacial drainage network affected ice flow and erosion? A lot of attention of the manuscript is focused on the overdeepenings themselves, but these slot canyons are also highlighted as a key feature so I would be interested to hear a bit more about the drainage system, if you feel confident enough to reconstruct it!

As a final note, while it is clear the authors have worked hard to integrate a lot of data for this paper, at times the manuscript does not fully reflect their work and can be a little sloppy. In the line-by-line comments, I highlight a few sentences where there are typos or the meaning of the sentence might be unclear. Additionally, I spent only a couple of minutes checking the citations and reference list and I noticed two or three mistakes already (that are listed in the minor comments section). I am sure there are other mistakes in the citations/reference list that I did not spot. Please can you thoroughly proof-read and double-check this for the next version of the manuscript.

**Minor comments, line by line comments:**

When I suggest any edits to a sentence, I use underline to indicate any suggested additions and  to indicate words that should be deleted.

Title: I'm not sure the purpose of 'tunnel valley' in the title. Is this to suggest that the whole Aare overdeepening is actually a tunnel valley incised by subglacial meltwater? I would probably remove '… or tunnel valley…' from the title.

Line 37: 'incision  by subglacial meltwater…'

I would probably also simply say deglaciation, rather than 'glacier's decay state'.

Line 67: 'a riegel is a rock wall' I would prefer a different term than rock wall, which gives the impression of a steep, exposed bedrock feature. Perhaps better to say 'bedrock bump oriented transverse to former glacier flow'.

Line 84: '… where we aimed  to exploring…'

Line 171: 'It bases on…' should be reworded to 'It is based on…'

Line 180: Which direction? Please can you specify the direction of the shallowing of the bedrock.

Line 318: citation reads 'Boultoon and Hindmarsch, 1987' but should read Boulton and Hindmarsh. The reference list has this correct.

Line 328: citation reads 'Hindmarsch', please change this to Hindmarsh.

Around line 350: You cite 'Stäger et al., 2020' multiple times but I cannot find this in the reference list.

Line 361: You do not need to define Lower Freshwater Molasse in this sentence as it has already been previously defined. Simply use the shortening of 'LFM'.

Line 421 – 423: This sentence is a little clunky. Reword to something like:

' Preusser et al. (2011) summarized multiple lines of evidence  that  piedmont glaciers  advanced into the Alpine foreland between 185 and 130 ka, i.e. the Beringen glaciation (MIS 6, Figure 8), and that this advance into the foreland was larger than during the LGM.'

Line 429 – 431: It is not immediately apparent how detailed mapping supports a chronological interpretation, please can you explain how the mapping evidence is inconsistent with an MIS 6 age? Even a brief explanation will make your assertion about the chronology more believable as opposed to vague statement about 'a-priori field-based information'. As someone without extensive regional knowledge I find statements like these unconvincing and just wishing I had more information!

I hope the authors find these comments useful and I apologize for the slight delay in preparing this review.

---

## Author Response (AR1)

*Dear Editor, dear Reviewers*

*We greatly acknowledge the detailed, careful and constructive reviews, which have been very helpful for us upon improving our paper. We considered all points indentified by the reviewers and updated the paper accordingly. The major changes include (i) an improvement of the structure and organization of the paper, (ii) a better description of the methodological approach, and (iii) the presentation of further data to constrain our reconstruction of the bedrock topography.*

*Please find below a point-by-point reply of how we handled the reviewers' comments and questions.*

*Pleae do not hesitate to contact me for further information.*

*On behalf of the co-authors*

*Fritz Schlunegger*

**Reviewer 1**

*Dear Reviewer*

*Many thanks for the very constructive and detailed comments on our manuscript. Please find below a reply of how we will handle the various concerns and questions.*

**Reviewer's comment:** While I agree with the majority of the explanation and argumentation that the authors provide, I am not convinced by the central premise of the manuscript, i.e. the existence of the bedrock riegel and slot canyons. The former is indicated by geophysical (gravimetric) data that show a relatively low negative anomaly interpreted as a relatively low depth to bedrock in the Bern city area. The latter are inferred from previous borehole logs that show a much greater drilled depth to bedrock. As far as I could gather from the manuscript, there is no drilling that would clearly support the existence of the riegel, as much as there is no geophysical evidence for the existence of the narrow gorges (due to limited resolution of the method). To me it appears not very likely that all deep boreholes in the area happened to strike narrow slot canyons in an otherwise massive bedrock riegel.

*We include a new Figure 4 showing the contour lines of equal Bouguer anomaly values, which we have calculated using the density of bedrock (2500 kg/m$^3$, Bandou et al., 2022) as reference density and not the 2670 kg/m$^3$ standard density value that is used per default for such corrections. The pattern of the contour lines will clearly document that the target overdeepening is generally broad and deep upstream of Bern, shallow beneath the city of Bern, and then narrow and deep downstream of it, similar to what we display in Figure 5a where we illustrate the pattern of the residual gravity anomalies related to the Quaternary sediments. This is used as a further piece of evidence that such a bedrock riegel exists beneath Bern. We also present, in a new Figure 2d, the logs of further drillings, which document that Quaternary sequences have been encountered in deep drillings beneath the city of Bern. In addition, we include the gravimetry profile of the Bern2 section (see Figure 2 in the submitted manuscript for location) as supplementary file, originally published in*

*Bandou et al. (2023), where a second residual gravity anomaly with a short wavelength has been identified by c. 10 gravity stations. We document that this secondary residual anomaly occurs beneath a large-wavelength residual gravity anomaly. Bandou et al. (2023) considered the anomaly with a large wavelength as the gravity signal of the bedrock riegel, whereas the short-wavelength anomaly beneath it illustrates the possible occurrence of a slot canyon, filled by Quaternary sediments. In summary, we are confronted with the situation that there is a bedrock riegel confirmed by several partly independent pieces of evidence, but that thick Quaternary deposits have been encountered in some deep drillings as well (and have also been detected in the Bern2 gravity profile). We thus prefer an interpretation where a bedrock riegel is cut by narrow slot canyons filled with Quaternary sediments (see other options below), as such a scenario adequately combines the findings from our gravity survey and drilling information. The bedrock riegel beneath Bern has indeed not been confirmed by a drilling. However, the shallow bedrock is situated beneath public park where deep drillings have not been conducted.*

**Reviewer's comment:** I would argue that a more straightforward explanation for these findings would be i) a local underestimation of the depth of the Quaternary infill by the gravimetric study

*We discard this hypothesis, because the regional Bouguer anomaly map where Bouguer corrections are accomplished with the bedrock's density clearly document that the bedrock is shallower in the city area of Bern than south and north of it – please see our reply above. We can rule out that the occurrence of Quaternary sediments with a much higher density (at least twice the density of the other Quaternary deposits) is responsible for this low gravity signal beneath Bern as this would invoke the occurrence of Quaternary sequences made up of a stack of e.g., highly compacted glacial till deposits. Such a scenario would offer an unrealistic situation that cannot not be combined with currently available information about the geologic architecture in the region (see log of Metas drilling in the new Figure 2d). Therefore, we consider this option as not really realistic.*

**Reviewer's comment:** or ii) an misinterpretation of some commercial drillings by the respective contractor that led to overestimation of the drilled bedrock depth.

*We present information from drillings, which clearly document that at some sites, the bedrock-Quaternary contact is very deep (new Figure 2d). It is possible that for some deep drillings, the contact between the Quaternary sediments and the Molasse bedrock has been misinterpreted by the respective contractors. However, we consider it unrealistic that this has been the case for nearly all drillings. Therefore we consider the option offered by the reviewer as not really realistic.*

**Reviewer's comment:** I would insist that these possibilities have to be at least seriously discussed in the manuscript. For i), a more detailed characterisation of the sedimentary infill of the overdeepening (especially its heterogeneity) might be very helpful.

*We have done this. We take the opportunity to include new figures (contour lines of Bouguer anomaly values where we used the bedrock density of 2500 kg/m³ for Bouguer corrections, logs of two drillings, and the residual gravity anomaly pattern of the Bern2 profile of Bandou et al., 2023) in the Supplement to sustain our original interpretation, and we included a new section in the discussion to further address this point.*

**Reviewer's comment:** Concerning the proposed scenario of riegel and canyons in the Aare main overdeepening, I further have two rather general questions. Addressing these could in my opinion strengthen the line of reasoning considerably: As nicely shown in the manuscript, riegels and dissecting gorges are a relatively common feature of glacial valleys, and likely to be found in overdeepenings as well. To my knowledge, such gorges occur characteristically in high-strength rocks like crystallines or limestones, and the examples mentioned in the text refer to such rock types too. Are there known examples of (subglacial) slot canyons in rather low-strength, moderately consolidated, porous rocks such as the Molasse sandstones in the Alpine foreland?

*We think that the downstream end of Lake Lucerne is such an example where the Upper Marine Molasse bedrock (the same unit as in the Bern area) forms a riegel that is cut by the transverse rivers. We include this example in our paper.*

**Reviewer's comment:** Are there known examples of anastomosing networks of slot canyons? I would imagine that the reactivation of a first, single incision through a riegel would always be 'easier' and thus more likely than the formation of a new incision, especially in such close proximity to the first that they might end up cross-cutting one another. Here, the relatively low erodibility of the Molasse sandstone could play a favourable role.

*We indeed consider the riegel/slot canyon ensemble near Meiringen to show such features (Figure 3a). We will outline this more clearly in the revised version of our manuscript. All other minor comments and suggestions will be considered upon drafting the revision of the paper.*

*Many thanks for the constructive review.*

*References: Bandou, D., Schlunegger, F., Kissling, E., Marti, U., Schwenk, M., Schläfli, P., Douillet, G., and Mair, Three-dimensional gravity modelling of a Quaternary overdeepening fill in the Bern area of Switzerland discloses two stages of glacial carving. Scientific Rep., 12, 1441, https://doi.org/10.1038/s41598-022-04830-x, 2022.*

*Bandou, D., Schlunegger, F., Kissling, E., Marti, U., Reber, R., and Pfander J.: Overdeepenings in the Swiss plateau: U-shaped geometries underlain by inner gorges. Swiss. J. Geosci., 116, 19, https://doi.org/10.1186/s00015-023-00447-y, 2023.*

**Citation**: https://doi.org/10.5194/egusphere-2024-683-AC1

**Reviewer 2**

*Dear Reviewer*

*Many thanks for the very constructive and detailed comments on our manuscript. Please find below a reply of how we will handle the various concerns and questions.*

**Reviewer's comment**: Reviewer 1 has already raised the issue of whether the interpreted riegel could be the result of either an underestimation of the Quaternary infill by gravimetry

and the misinterpretation of drillings and I note that you have already responded to these comments. So I don't want to dwell on this topic too much. But I welcome the introduction of new data that would support your interpretation and while I don't necessarily disagree with your interpretation, I would appreciate a brief discussion of the possible alternative interpretations to highlight some of the uncertainties and be more transparent to the reader. So I look forward to reading this addition.

*This has been done, and additional material is presented to support our interpretation. Please also see our response to the previous review.*

**Reviewer's comment**: On Figure 7, I would appreciate the location of deep drillings being labelled (in a similar style to Figure 6). This would allow the reader to easily see the ground-truthed evidence supporting the relatively low Quaternary infill (if any boreholes do show this) and location of slot canyons. While this information can be found in Figure 5, it can be quite difficult to interpret this data from a map figure and I would appreciate it plotted onto the cross-profiles as well.

*This has been done.*

**Reviewer's comment**: For example, on the Kehrsatz cross-profile you reconstruct a single slot canyon but on Figure 5 there don't appear to be any drillings which would support this reconstruction? If there is a drilling that supports it then could you label it on Figure 7c, even if the drilling is one of those located a few hundred metres up or downstream of the drawn cross-profile.

*We project one deep drilling situated between the Kehrsatz and Airport profiles onto these profiles. However, this has to be considered with caution because the projection distance is quite large (nearly 1 km).*

**Reviewer's comment**: Section 5.5 Chronological framework. A synthesis of dating evidence is mentioned in both the abstract and introduction of the paper, so I expected the discussion of the chronological constraints to be more important to the paper. But, this section feels a little bit like a brief review of the existing constraints and doesn't really significantly add anything new or contribute to your aim of exploring the origin of these overdeepenings. While it is important to understand the timing of glaciations that formed these overdeepenings, I believe that this section could be simplified and cut to about half it's current length. In its current length I felt like it distracted from the main message of the manuscript. In your conclusions section, you barely mention the chronology and I think this illustrates how this section isn't a key element of your manuscript and it's location in the discussion section is taking away from the message. Another alternative to shortening this section could be moving it earlier in the manuscript, perhaps after Section 2 on 'Riegels and slot canyons in the Alpine valleys…'

*We indeed agree. We shortened this section as suggested and include it into the description of the setting as suggested.*

**Reviewer's comment**: The evolution of the slot canyon network across the bedrock riegel. Across Figure 7, there appears to be an evolution of the subglacial drainage network that incised these slots canyons as it passes over the riegel. At the Aare cross-section, upstream of the riegel there appears to be no evidence of a channelised subglacial drainage network

incising into the bedrock (distributed subglacial drainage network or just no drilling data which identified a channel?). As you pass over the riegel there is the anastomosing network of slot canyons that you reconstruct. Then downstream of the riegel on the Bremgarten cross-profile, there is a single slot canyon (possibly?). I am aware that these reconstructed networks are very dependent on drilling data which struck one of these canyons and so this reconstruction has quite large uncertainties, but, I would be interested to see a description of this evolution of the channel network. If you believe that your data is reliable enough to reconstruct it, if not feel free to ignore this.

What are the processes that could drive these changes in network and how it relates to the changing bed slope? Additionally, how could these variations in the subglacial drainage network affected ice flow and erosion? A lot of attention of the manuscript is focused on the overdeepenings themselves, but these slot canyons are also highlighted as a key feature so I would be interested to hear a bit more about the drainage system, if you feel confident enough to reconstruct it!

*Our database is too sparse to enter into such a discussion. We would need further detailed information about the ages and the sedimentary architecture of the Quaternary fill. We indeed have no data to constrain these boundary conditions and therefore decided not to start such a discussion (which would be highly speculative), but we add a related note at the end of the paper.*

**Reviewer's comment**: As a final note, while it is clear the authors have worked hard to integrate a lot of data for this paper, at times the manuscript does not fully reflect their work and can be a little sloppy. In the line-by-line comments, I highlight a few sentences where there are typos or the meaning of the sentence might be unclear. Additionally, I spent only a couple of minutes checking the citations and reference list and I noticed two or three mistakes already (that are listed in the minor comments section). I am sure there are other mistakes in the citations/reference list that I did not spot. Please can you thoroughly proof-read and double-check this for the next version of the manuscript.

*Thanks for noting. We carefully checked the revised manuscript of typos, errors and correct citations.*

**Reviewer's comment**: Minor comments, line by line comments:

When I suggest any edits to a sentence, I use underline to indicate any suggested additions and strikethrough to indicate words that should be deleted.

Title: I'm not sure the purpose of 'tunnel valley' in the title. Is this to suggest that the whole Aare overdeepening is actually a tunnel valley incised by subglacial meltwater? I would probably remove '… or tunnel valley…' from the title.

*Thanks for suggesting. We have changed the title accordingly.*

**Reviewer's comment**: Line 37: 'incision of by subglacial meltwater…'

*Corrected*

**Reviewer's comment**: I would probably also simply say deglaciation, rather than 'glacier's decay state'.

*Yes indeed; we have corrected the text accordingly.*

**Reviewer's comment**: Line 67: 'a riegel is a rock wall' I would prefer a different term than rock wall, which gives the impression of a steep, exposed bedrock feature. Perhaps better to say 'bedrock bump oriented transverse to former glacier flow'.

*Thanks for the suggestion; we have corrected the text accordingly.*

**Reviewer's comment**: Line 84: '… where we aimed at to exploring…'

*…'we aimed at exploring' is correct use of English. We have checked it.*

**Reviewer's comment**: Line 171: 'It bases on…' should be reworded to 'It is based on…'

*Corrected.*

**Reviewer's comment**: Line 180: Which direction? Please can you specify the direction of the shallowing of the bedrock.

*We refer to the orientation and have thus changed the text accordingly.*

**Reviewer's comment**: Line 318: citation reads 'Boultoon and Hindmarsch, 1987' but should read Boulton and Hindmarsh. The reference list has this correct.

*Corrected.*

**Reviewer's comment**: Line 328: citation reads 'Hindmarsch', please change this to Hindmarsh.

*Thanks for noting. This has been corrected.*

**Reviewer's comment**: Around line 350: You cite 'Stäger et al., 2020' multiple times but I cannot find this in the reference list.

*Corrected.*

**Reviewer's comment**: Line 361: You do not need to define Lower Freshwater Molasse in this sentence as it has already been previously defined. Simply use the shortening of 'LFM'.

*Corrected.*

**Reviewer's comment**: Line 421 – 423: This sentence is a little clunky. Reword to something like: 'Yet, Preusser et al. (2011) summarized multiple lines of evidence for proposing that the piedmont glaciers did advanced into the Alpine foreland between 185 and 130 ka, i.e. the Beringen glaciation (MIS 6, Figure 8), and that this advance into the foreland was larger than during the LGM.'

*We have removed this sentence upon shortening the related chapter.*

**Reviewer's comment**: Line 429 – 431: It is not immediately apparent how detailed mapping supports a chronological interpretation, please can you explain how the mapping evidence is inconsistent with an MIS 6 age? Even a brief explanation will make your assertion about the chronology more believable as opposed to vague statement about 'a-priori field-based information'. As someone without extensive regional knowledge I find statements like these unconvincing and just wishing I had more information!

*We have removed this sentence upon shortening the related chapter.*

**Reviewer's comment**: I hope the authors find these comments useful and I apologize for the slight delay in preparing this review.

*Thank you very much! They are very helpful.*

---

## Author Response (AR2)

*Dear Editor, dear Reviewers*
*We greatly acknowledge the detailed, careful and constructive review. Please find below a reply of how we handled the reviewer's comments and questions. Please note that we took the opportunity to correct the coordinate framework of Figure 6. The original coordinates were offset as the framework was shifted by a few hundreds of meters. We have now corrected this pitfall.*
*Please do not hesitate to contact me for further information.*

*On behalf of the co-authors*
*Fritz Schlunegger*

**Anonymous reviewer comment**
The revision has addressed the reviewers' recommendations well, and has improved the manuscript considerably. I appreciate that more sedimentological and chronological context is provided, and that some additional, important aspects have been addressed in the discussion.

What is striking me, now that the drill logs have been added, is the difference in sediment infill between Marzili and Metas (where the riegel is proposed), and at Brunnenbohrung and Rehhag. At the latter two sites it consists to a large portion of lacustrine fines. consist to a large portion of lacustrine fines. At the former it is is described as 'mud and sand layers, which contain isolated clasts' but shown as 'gravel with mud' in the log. I think it is important to be as precise as possible here, as gravelly sediments can have bulk densities as high as the bedrock density (e.g. https://doi.org/10.5194/sd-32-27-2023, https://doi.org/10.5194/sd-33-191-2024). I suggest to harmonize the sediment description in text and drill log,

*This has been done. Thanks for this suggestion. We additionally added the log of a further drilling to Figure 2d.*

…. and to discuss what kind of impact these sedimentological differences might have on the inferred valley morphology.

*This is indeed the case, and we further discuss this point in section 2.5 of the revised text. However, based on 3D gravity modelling (Bandou et al., 2023; Swiss J. Geosci.), we found that the bulk densities of the Quaternary sediments depend less on the lithological architecture of the material or the depositional environment in which the sediments were deposited. Instead, they are primarily a function of the maximum depth of the overdeepening fill and the number of glaciations, during which the Quaternary sediments were compacted under a thick glacial cover. For instance, a sequence postdating MIS 6 was compacted by the piedmont glacier during the Last Glacial Maxiumum (LGM) during MIS 2 only, while older sediments experienced a glacial compaction during at least two full glaciations. We thus expanded the revised text accordingly.*

If this last point can be addressed by the authors, I don't see anything opposing publication of the paper.
*Thank you very much for the careful review.*